# Reliability of ultrasonographic measurement of muscle architecture of the gastrocnemius medialis and gastrocnemius lateralis

**Samantha May** [1¤], **Simon Locke**[1], **Michael Kingsley** [2,3]*

1 La Trobe Rural Health School, La Trobe University, Bendigo, Victoria, Australia, 2 Holsworth Research Initiative, La Trobe University, Bendigo, Victoria, Australia, 3 Department of Exercise Sciences, University of Auckland, Auckland, New Zealand

¤ Current address: Department of Sports Medicine, Alphington Sports Medicine Clinic, Northcote, Victoria, Australia

* M.Kingsley@latrobe.edu.au, Michael.Kingsley@Auckland.ac.nz

**Data Availability Statement:** The data underlying the results presented in the study are available from: https://doi.org/10.26181/611a2abef0c7e.

## Abstract

Ultrasonography is widely used to measure gastrocnemius muscle architecture; however, it is unclear if values obtained from digitised images are sensitive enough to track architectural responses to clinical interventions. The purpose of this study was to explore the reliability and determine the minimal detectable change (MDC) of gastrocnemius medialis (GM) and gastrocnemius lateralis (GL) muscle architecture using ultrasound in a clinical setting. A trained sonographer obtained three B-mode images from each of the GM and GL muscles in 87 volunteers (44 males, 43 females; 22±9 years of age) on two separate occasions. Three independent investigators received training, then digitised the images to determine intra-rater, inter-rater, and test-retest reliability for fascicle length (FL), pennation angle (θ) and muscle thickness. Median FL, θ, and muscle thickness for GM and GL were 53.6–55.7 mm and 65.8–69.3 mm, 18.7–19.5˚ and 11.9–12.5˚, and 12.8–13.2 mm and 15.9–16.9 mm, respectively. Intra- and inter-rater reliability of manual digitisation was excellent for all parameters. Test-retest reliability was moderate to excellent with intraclass correlation coefficient (ICC) values ≥0.80 for FL, ≥0.61 for θ, and ≥0.81 for muscle thickness, in both GM and GL. The respective MDC for GM and GL FL, θ, and muscle thickness was ≤12.1 mm and ≤18.00 mm, ≤6.4˚ and ≤4.2˚, and ≤3.2 mm and ≤3.1 mm. Although reliable, the relatively large MDC suggest that clinically derived ultrasound measurements of muscle architecture in GM and GL are more likely to be useful to detect differences between populations than to detect changes in muscle architecture following interventions.

## Introduction

Skeletal muscle architecture is described by the geometrical arrangement of muscle fibre bundles relative to the force-generating axis of the muscle [1]. The arrangement influences contraction velocity and force-generating capacity of a muscle, as well as the range of reciprocating motion, known as excursion [1–5]. In pennate muscles, fibre bundles known as

**Funding:** The authors received no specific funding for this work.

**Competing interests:** The authors have declared that no competing interests exist.

fascicles, insert obliquely into aponeuroses. The angle derived from the insertion of a fascicle into an aponeurosis defines the pennation angle (θ), and the distance between aponeuroses defines anatomical muscle thickness. These parameters of skeletal muscle architecture have been measured in studies of muscle physiology and biomechanics to determine anatomical and contractile properties of muscles [2, 3, 6–10]. Fascicle length (FL), θ, and muscle thickness, can be measured through in vivo, two-dimensional (2D) B-mode ultrasound imaging [11–14]. B-Mode ultrasound is non-invasive, readily available in research and clinical settings, and provides superior spatial resolution of images obtained in real time. This modality provides detailed visualisations of hypoechoic fibre bundles with hyperechoic septations within the muscle, separated by hyperechoic fibroadipose tissue [3, 7, 8, 15–18]. Once a single image frame or set of images are acquired, measurements are traditionally made using manual digitisation with custom-written computer software [2, 8, 19–27] or by using automatic tracking [28–31].

Several limitations should be considered when using ultrasound to measure skeletal muscle architecture. Ultrasound is operator-dependent, and potential sources of error stem from probe placement or location, probe pressure, and probe orientation [32, 33]. Freehand probe placement allows for manual adjustments of the probe parallel to the fascicle plane to obtain the clearest possible image, although this 'field-based' approach is subject to operator bias and can reduce reliability of repeated measurements on the same subject [2, 32–38]. Misalignment of the probe plane with respect to the fascicle plane can lead to over- or under-estimation of FL and θ [32, 37]. Excessive probe pressure can lead to tissue deformation and reduce the accuracy of measurements [32]. Standard configurations, which are most cost-effective, include a transducer sized between 4 to 6 cm. These transducer field of view dimensions are often too narrow to simultaneously view both a fascicle's origin on one aponeurosis and its insertion on the other. Specialised long transducers (10 cm) can be employed; however, these transducers are more costly and increase susceptibility to uneven probe pressure spread across the underlying tissues that can cause deformation [39, 40]. More advanced extended-field-of-view techniques, where entire fascicles are imaged by moving the probe with a continuous scan, increase the risk of technical error during digital reconstruction of the image [41]. In cases where the above techniques are not available and fascicles extend beyond the ultrasound's field of view, the external segment can be estimated by linear extrapolation of the fascicle to its intersection with the aponeuroses [2, 24, 25, 42]. Linear extrapolation assumes a fascicle follows a linear path and does not account for fascicle or aponeurosis curvature, which has important implications for the accurate calculation of architecture [43, 44]. Consequently, the reproducibility of conventional B-mode ultrasound measures of skeletal muscle architecture requires investigation.

Authors of a systematic review found that, with appropriate operator training, ultrasound measures of muscle architecture are highly reliable and valid [11]. However, most of the studies in the review described strict protocols to reduce operator bias and error, used expensive machines, or used equipment that cannot be transported to other relevant sites where sports medicine services are engaged, such as athletic clubrooms [45]. The procedures that standardise: (1) positioning of the subject and ultrasound transducer, (2) scan parameters, (3) image processing, and (4) digitisation analysis are time consuming, which poses a challenge to the suitability of the use of ultrasound in a clinical setting where time and equipment are often limited [40]. Therefore, the reproducibility of findings in a controlled environment might not translate to a clinical setting, where high resolution stationary ultrasound machines or retest aids are not universally available. Exploration of the reproducibility of an ultrasound method applicable to a clinical sports medicine context has not been performed. Regardless of this,

conventional B-mode ultrasound imaging remains the most widely used modality to measure skeletal muscle architecture in clinical settings due to its practicality and affordability [39].

The gastrocnemius medialis (GM) muscle architecture has been determined to relate closely to force production capacity and ankle joint kinematics during gait [25]. The muscle is located superficially relative to skin [3, 11], and contains shorter fascicles that can be measured with a single commercial transducer sized 4–6 cm. Because of the GM's locomotor significance and ease of identification of muscle architecture under ultrasound, it has been frequently targeted in interventional studies aimed at improving biomechanical function, as well as in clinical studies of muscle disorders [46], and immobilisation studies exploring bed rest [25] or exposure to microgravity [47]. The GM combines with the gastrocnemius lateralis (GL) as well as the soleus combine to form the triceps surae muscle group, the dominant plantar flexors of the ankle [48]. The gastrocnemius muscles have different architectural properties; GL is characterised by longer FLs and smaller θs whereas the GM comprises shorter FLs and larger θs [7, 47].

The reliability of measuring GM architecture using ultrasonography has been reported as being between good to excellent for FL (ICC = 0.81–0.99) and θ (ICC = 0.85–1.00) following a review of studies [11] where researchers applied strict protocols with step-by-step instructions [2, 8, 24, 25, 42, 43, 46, 49]. Across these studies, different measures were repeated for the purposes of reporting reliability, and sample sizes were relatively small, which might compromise the conclusions that can be drawn. Authors analysed the reliability of image acquisition, image analysis, or both of these steps with repeated measurements, however these were performed either within a single session, across multiple sessions, or across multiple investigators [11]. These variations have made it difficult for inter-study comparisons of the values reported. When simplified to the step of image analysis alone, specifically the method of manual digitisation, studies have reported excellent reliability following a single investigator repeating the digitisation step on two occasions (intra-rater), or between multiple investigators (inter-rater) [2, 8, 11, 24, 25, 32, 42, 43, 46, 49]. However, when investigating responses to interventions, it is more valuable to determine whether the method of both image acquisition and digitisation repeated on multiple occasions is reliable and sensitive enough to track adaptations.

Historically, GM measurements are more vulnerable to error when measured between sessions [11]. Two studies revealed large discrepancies in GM FL and θ results after being scanned with ultrasound twice, and recommended caution regarding small changes in FL and θ in clinical studies as the relevant changes were likely smaller than the measurement error [32, 50]. Existing literature does not address whether the magnitude of measurement error of ultrasonography exceeds the potential changes in FL, θ, and muscle thickness following clinical interventions such as immobilisation, chronic exercise training, or disuse. The reliability of imaging the GL is far less known [11], and it cannot be assumed that the method of ultrasound imaging for GM will prove reliable for GL in the same individual or group [7, 51]. Separate analysis is required, particularly test-retest reliability estimates including the minimum detectable change. This new knowledge can inform the appropriateness of tracking architectural adaptations to both heads of the gastrocnemius muscle in individuals using ultrasound.

The primary purpose of this study was twofold: (1) to determine the reliability of ultrasound image acquisition and manual digitisation of GM and GL muscle architecture parameters in resting conditions in a clinical setting, and (2) to estimate the magnitude of difference that would be required to detect real change between or within individuals and groups. The authors hypothesised that: (a) the intra-rater and inter-rater reliability of the manual digitisation step would be excellent and align with results from previous studies, and (b) that the test-retest reproducibility of the muscle architecture results between ultrasound sessions using methods repeated in a clinical setting would be lower than previously described results in the literature

that applied stricter conditions or used equipment not readily available to the sports medicine clinician.

## Methods

### Study design and participants

Fifty-six adults and 31 adolescents aged between 13 and 63 (44 males and 43 females: mean ± SD, 22 ± 9 years, 175 ± 12 cm, and 73 ± 16 kg) provided signed consent and attended two data collection sessions at least 7 days apart. An additional 2 adult and 4 adolescent participants attended a single data collection session but did not attend the second data collection session and were not included in the test-retest reliability analysis. For participants aged <18 years, written informed consent was given by a parent or legal guardian. Participants were screened for eligibility using a questionnaire. The questionnaire covered demographic information, such as age, gender, previous lower limb injuries, and activity levels. Participants were excluded if they were aged 12 or younger, had a history of an acute or chronic lower limb injury within the previous 12 months, or if they had exercised their calf muscles earlier on the day of the assessment (prolonged walking, jogging, running, sprinting, hopping, skipping, jumping, or performing heel raises). The study was approved by La Trobe University Human Ethics Committee (Reference: S17-114) and was conducted according to the National Statement on Ethical Conduct in Human Research.

### Procedures

Anthropometry was undertaken in accordance with procedures described by the International Society for the Advancement of Kinanthropometry [52]. Body mass was recorded to the nearest 0.1 kg using a portable analogue floor-scale (Model 762; Seca, Germany), and stretch stature was measured to the nearest 0.1 cm using a portable stadiometer (Model 213; Seca; Germany). The muscle architecture of both left and right GM and GL muscles were assessed during resting conditions to reduce fascicle curvature [2, 43]. B-mode ultrasound images were taken from each participant on two occasions using a portable ultrasound (LOGIQ V2; GE Healthcare, Australia) with a 38 mm wide linear probe and a standardised frequency of 12–13 MHz. This portable machine was selected as this is attractive for clinical use given its lower cost and smaller size compared to large, stationary, high-precision machines [53].

Participants lay prone on an examination table with the lower leg supported on an inclined foam wedge angled at 20˚ so that the knee was flexed to within the desired range of 20–30˚ [54]. The wedge was consistently placed in the same position with the thin edge adjacent to the tibiofemoral joint lines to ensure the retest position would be maintained; the knee angle was confirmed on the first 20 participants with a manual goniometer. The gastrocnemius bridges both the knee and ankle joint and is under full tension when the knee is extended because the muscle origin is furthest from its insertion [54]. Knee flexion of at least 20˚ eliminates the effect of ankle dorsiflexion restraining the gastrocnemius, and ankle dorsiflexion and gastrocnemius tension remains unchanged between 20˚ and 75˚ of knee flexion [54]. The ankle was secured at approximately 90˚ using a night splint and confirmed with a manual goniometer (Fig 1). A trained sonographer used a measuring tape to find the initial probe site at one-third of the distance from the popliteal crease of the knee to the tip of the medial malleolus for the GM and the lateral malleolus for the GL, at the mid-muscle belly which was determined via inspection and palpation [7, 27, 32, 47, 49, 53, 55]. The sonographer applied ample conductive gel to ensure acoustic enhancement and reduce probe pressure against the skin. The probe was then orientated by freehand to the plane of fascicles that were visibly inserting into either the superficial or deep aponeurosis, and where the aponeuroses were close to parallel [23, 33, 37].

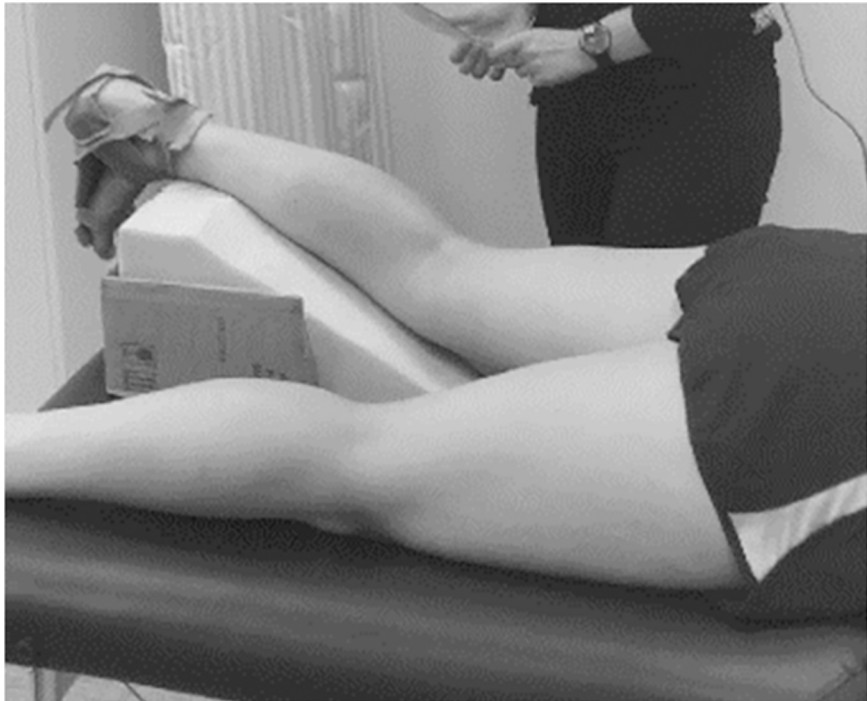

**Fig 1. Position of lower limb during ultrasonography.**

This positioning around the initial placement site appreciates learned clinical practice where a structure is targeted and visualised in real time using pattern recognition [56, 57]. Depth of scanning was manually adjusted to 3–6 cm from the surface of the transducer, until both outer borders of the superficial and deep aponeuroses were visible. Multiple ultrasound images were obtained at the site where the muscle belly was widest, where FLs are thought to be homogenously distributed [8, 58]. The probe was lifted between image captures to ensure each muscle's architecture was represented by several different measures of FL, θ and muscle thickness. This resulted in three ultrasound images captured from each GM and GL at the left and right limb on the same day within one session, totalling 12 images per participant. This process was repeated at a second session held at least 7 days later, resulting in a total of 24 images captured per participant.

Images underwent de-identification and randomisation (https://www.randomizer.org). The following anatomical structures were digitised frame-by-frame (Fig 2), using software designed in LabVIEW (version 16; National Instruments, USA): (1) superficial aponeurosis external and internal borders; (2) deep aponeurosis external and internal borders; and (3) proximal and distal endpoints of three muscle fascicles within the imaging plane per image. FL was defined by the length of a linear fascicle path between the superficial aponeurotic junction and the deep aponeurotic junction. Investigators received one hour of training and were provided with step-by-step training manuals that instructed investigators to prioritise selection of fascicles that were straight with visible endpoints at the junction of either the deep or superficial aponeurosis. Where the fascicles extended beyond the captured image, the software extrapolated to the aponeuroses and estimated FL, like that described in model V by Ando et al. [59]. The intersection of the line of the selected fascicle inserting obliquely with the deep aponeurosis defined θ. An average of these three measurements was used to represent FL and θ [60]. Muscle thickness was defined as the mean vertical distance between the internal borders

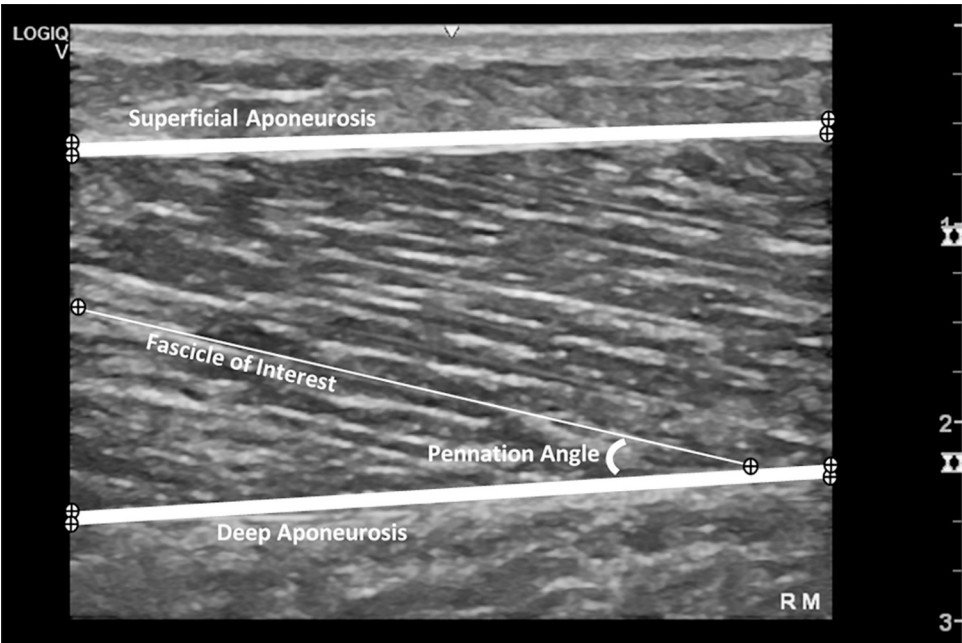

**Fig 2. Ultrasound image from a right medial head of gastrocnemius of a participant, showing an example of the points plotted (⊕) for superficial aponeurosis borders, deep aponeurosis borders, and a selected fascicle length and corresponding pennation angle.**

of the superficial and deep aponeuroses measured on either side of the image. One investigator reviewed all 2184 de-identified images and removed 39 images (1.7%) due to poor quality, determined by inability to visualise the borders of deep aponeurosis (incorrect depth), or an unacceptable degree of curvature in the aponeuroses.

Fig 3 describes the methods used to determine reliability for the steps of the procedure. Intra-rater reliability of the digitisation method was analysed using one blinded investigator (Investigator A) repeating digitisation on 100 randomly sampled images on two occasions. Three blinded investigators (A, B and C) analysed all 2145 images to provide comparisons for inter-rater reliability of the digitisation method. The images were divided into images of GM (1081 images) or GL (1064 images), and reliability analyses were performed for these groups separately. For test-retest reliability of the ultrasound image acquisition and digitisation method combined, results were re-identified for each of the 87 participants and separated into

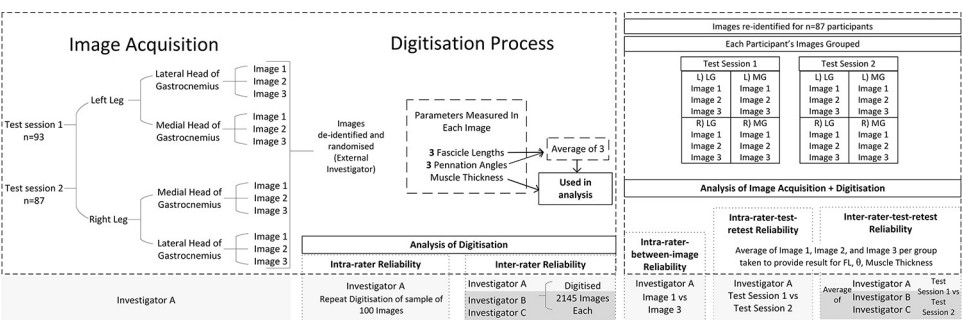

**Fig 3. Protocols for reliability analyses.**

image number (1, 2 or 3) and test session (1 or 2). Intra-rater between-image reliability was measured by comparing Investigator A's results from session 1 image 1, to session 1 image 3. Intra-rater-test-retest reliability was determined by comparing Investigator A's averages from test session 1 images to test session 2. The average of the three investigators (A, B and C) results from test session 1 was also compared against their average of results from test session 2 to determine inter-rater-test-retest reliability.

## Statistical analysis

FL, θ, and muscle thickness were assessed for normality using the Shapiro-Wilk test, alongside assessment of skewness, kurtosis, and distribution curves. The majority of data breached the assumption of normality ($p < 0.05$), and group data are presented as median and inter-quartile range (IQR).

Relative consistency was quantified using intraclass correlation coefficients (ICC) according to guidelines defined by McGraw and Wong (1996) with 95% confidence intervals (95% CI) [61]. The ICC values were calculated for intra-rater (ICC; two-way mixed-effects model, absolute definition, single rater type), inter-rater (ICC; two-way random-effects model, absolute definition, single rater type), and test-retest reproducibility (two-way mixed-effects model, absolute definition, average rater type) [61, 62]. ICC values were classified according to Koo and Li (2016) as excellent ($\geq$0.90), good (0.75 to 0.89) moderate (0.50 to 0.74), and poor ($\leq$0.49) [62]. Absolute reliability was determined using the methods of Bland and Altman (1999) [63]. Heteroscedasticity was examined through visual inspection of the Bland-Altman plots and the correlation between the absolute differences and their respective means. The correlation was significant for the majority of comparisons, and data were log transformed prior to further analysis of the agreement between measures [63]. Consequently, mean bias and 95% ratio limits of agreement (RLOA) relative to the mean were calculated to provide an indication of systematic error and random error, respectively. These values show how much two scores varied between investigators or between sessions; mean bias values closer to 0 indicate higher reliability, and narrower RLOA are preferable [64].

Test-retest reliability was further assessed by calculating the standard error of the measurement (SEM), as follows: $SEM = SD\sqrt{1 - ICC}$, where SD is the standard deviation of the measure [65]. The minimal detectable change (MDC) defines the smallest change in the scores between two sessions that needs to be observed in order to be 95% confident that the observed change in muscle architecture is real and not a product of measurement error or random variation in the method [66]. The SEM was used to calculate the MDC with the equation: $MDC = SEM \times \sqrt{2} \times z\ score_{(95\%\ CI)}$, where the *z score* equals 1.96, and the multiplier of $\sqrt{2}$ is to account for the uncertainty introduced by using difference scores from measurements at 2 points in time [67, 68]. In addition, Cohen's d effect size, measures the proportion of variance shared by groups, and was calculated as the difference in group mean values divided by the square root of the pooled variance (population SD) [69]. An effect size of 0.20 was considered to be small, 0.50 medium, and 0.80 large [70]. Results were considered significant at $p \leq 0.05$, and statistical analyses were performed using IBM SPSS (V25.0; Armonk, NY: IBM Corp.) [71].

## Results

### Analysis of manual digitisation

**Intra-rater reliability.** Intra-rater reliability was excellent for FL (ICC = 0.99, 95% CI: 0.98–0.99), θ (ICC = 0.99, 95% CI: 0.98–0.99), and for muscle thickness (ICC = 0.99, 95% CI:

**Table 1. Gastrocnemius medialis and gastrocnemius lateralis median (IQR) values and inter-rater reliability results for fascicle length, pennation angle and muscle thickness lengths.**

| Gastrocnemius Medialis (1081 images) | | | | Gastrocnemius Lateralis (1064 images) | | | |
|---|---|---|---|---|---|---|---|
| Median (IQR) | ICC (95% CI) | Mean Bias (RLOA) | Cohens | Median (IQR) | ICC (95% CI) | Mean Bias (RLOA) | Cohens |
| *Fascicle Length* | | | | | | | |
| 54.6 mm (49.1–62.9) | 0.95 (0.95 to 0.96) | -0.5% (-12.3 to 12.8) | 0.07 | 67.6 mm (59.6 to 77.3) | 0.91 (0.90 to 0.92) | -0.8% (-16.5 to 17.7) | 0.08 |
| *Pennation Angle* | | | | | | | |
| 19.1˚ (16.4–21.5) | 0.95 (0.94 to 0.96) | 0.3% (-11.4 to 13.5) | 0.05 | 12.2˚ (10.1 to 13.9) | 0.94 (0.94 to 0.95) | 1.5% (-13.6 to 19.2) | 0.18 |
| *Thickness* | | | | | | | |
| 16.4 mm (14.6–18.7) | 1.00 (1.00 to 1.00) | 0.1% (-3.1 to 3.4) | 0.07 | 13.0 mm (11.0 to 14.8) | 0.99 (0.99 to 1.00) | 0.2% (-3.9 to 4.5) | 0.11 |

IQR = interquartile range; ICC = intra-class correlation coefficient; CI = confidence interval; RLOA = ratio limits of agreement, 95% CI; mm = millimetres; ˚ = degrees.

0.99–0.99). Systematic bias was small, with mean differences (MD) of 0.3%, 0.7% and 0.1%, respectively. In absolute values, MD were 0.3 mm, 0.1˚, and 0.1 mm. The RLOA were -8.1% to 8.0% for FL, -9.4% to 11.9% for θ, and -2.7% to 3.0% for muscle thickness.

**Inter-rater reliability.** Table 1 displays median (IQR) values and inter-rater reliability analyses for the acquired images of GM and GL. For all comparisons, the ICC values were excellent ($\geq 0.91$), with the widest 95% CI being 0.90–0.92. The ICC values were strongest for muscle thickness comparisons, with ICC values of 0.99 to 1.00. Mean differences between Investigators A, B and C were small in magnitude for both GM and GL. For all parameters, there was no evidence of systematic bias (Fig 4), and the mean bias ranged from 0.1% to 0.5% for GM and 0.2% to 1.5% for GL. For example, the mean bias for GM FL equates to small absolute values ($\leq 0.3$ mm) in proportion to the median FL of 54.6 mm. For GL FL, the difference in absolute values was again small ($\leq 0.5$ mm) against a median FL of 67.6 mm. The GL θ showed the largest systematic difference which was found between Investigators A and B: MD: 2.2% (RLOA: -12.1%– 18.7%), as well as between Investigators A and C being MD: 2.2% (RLOA: -13.3%– 20.5%) (S1 Table). This equates to 0.1˚ for both, corresponding to 0.8% of the median angle. For both muscles, muscle thickness was the most reliable parameter measured with the tightest RLOA (-3.1% to 3.4% for GM and -3.9% to 4.5% for GL). Cohens d statistic was small ($< 0.20$) in all comparisons.

## Analysis of image acquisition and manual digitisation

**Intra-rater between image reliability.** Relative reliability was excellent for FL (ICC = 0.95, 95% CI: 0.94–0.96), θ (ICC = 0.94, 95% CI: 0.93–0.95), and for muscle thickness (ICC = 0.97, 95% CI: 0.97–0.98). Systematic bias between image 1 to image 3 was small, with mean differences (MD) of 0.2%, 1.1% and 0.1% respectively. The RLOA were wide, within the range of -14.9% to 17.2% for FL, -19.1% to 26.2% for θ, and 10.0% to 10.8% for muscle thickness.

**Test-retest reliability.** The results of the image acquisition and digitisation inter-rater-test-retest and intra-rater-test-retest analyses are presented in Table 2. GM inter-rater and intra-rater comparisons for FL was the most reliable measure, with good to excellent relative reliability (ICC range: 0.88 to 0.92). ICCs were moderate to good for θs with the widest values ranging from 0.63 to 0.76, and good for muscle thickness (ICC range: 0.83 to 0.86). Similar findings were seen for GL (Table 2); FL was less reliable (ICC range: 0.80 to 0.84). Mean results and MDC values were similar between intra-rater and inter-rater analyses. Mean bias was $\leq 3.6\%$ for all parameter comparisons in both muscles, and there were no systematic differences between the left limb and right limb. The left limb results are represented in the

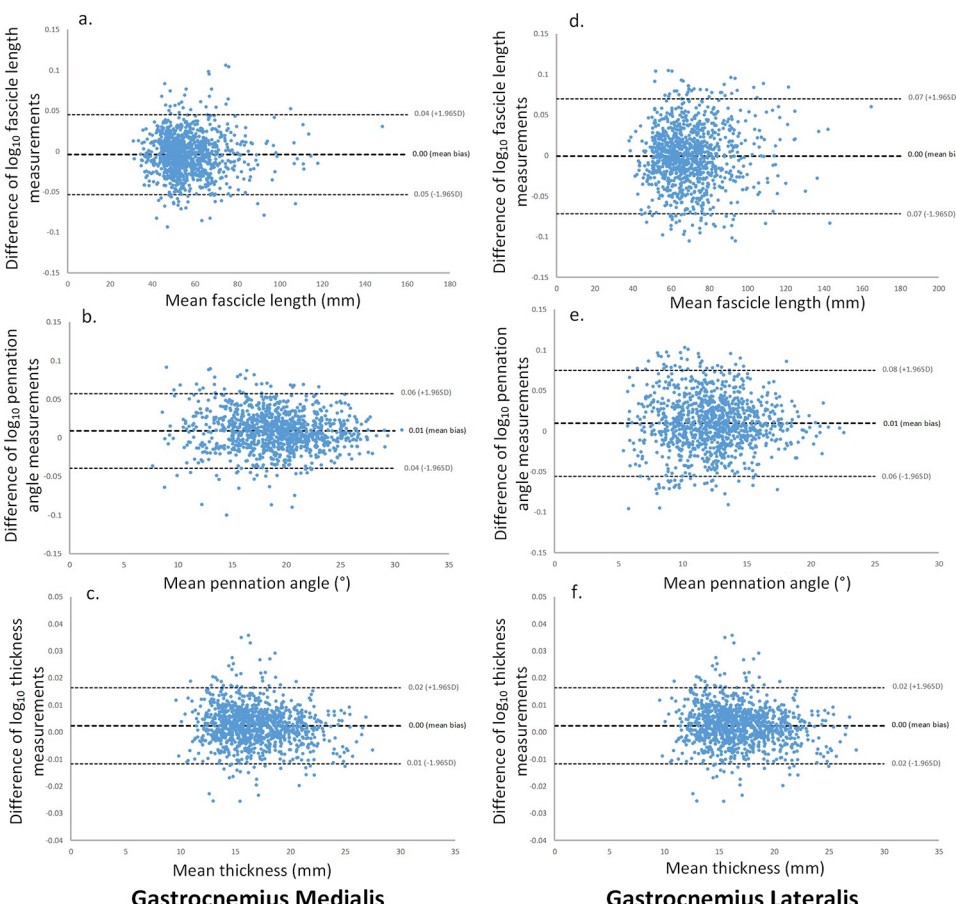

**Gastrocnemius Medialis**          **Gastrocnemius Lateralis**

**Fig 4. Bland-Altman plots showing the differences between Investigator A and Investigator B log10 measurements in relation to the mean of Investigator A and Investigator B measurements.** Dashed lines indicate the mean bias and the upper and lower 95% limits of agreement. Fascicle length, pennation angle, and muscle thickness measurement comparisons for the gastrocnemius medialis are represented by a. b. and c. respectively, and the gastrocnemius lateralis represented by d. e. and f.

Bland-Altman plots in Fig 5. The mean bias was close to zero for all parameters. In contrast, the 95% RLOA were wide, demonstrating random error in the method. In terms of precision, absolute reliability was stronger when measuring muscle thickness in both the GM (SEM = 1.1–1.2; MDC = 3.2 mm) and the GL muscles (SEM = 0.9–1.6; MDC = 2.6–3.1 mm). The largest inter-rater GM FL MDC was 12.1 mm, which represents 22.0% the median 54.9 mm. For the GL, the greatest inter-rater FL MDC was 17.3 mm, or 25.9% of the associated FL median of 66.7 mm. Cohens values were < 0.29 for all parameter comparisons.

## Discussion

The findings confirm that reliability is excellent when using manual digitisation to measure gastrocnemius FL, θ and muscle thickness on images derived from ultrasound, within a single investigator on two occasions, and between multiple investigators (ICCs ≥ 0.91). Inter-rater digitisation results for the GL were slightly less reliable when compared to the GM, likely due to the anatomical differences between the two muscles. Reliability is reduced when ultrasound image acquisition is repeated within a session which required the probe to be removed and replaced between scans. Reliability is further reduced when participants attend re-scanning on

**Table 2. Test-retest reliability for fascicle length, pennation angle and muscle thickness of the gastrocnemius medialis and gastrocnemius lateralis muscles.**

| | Test1 Median (IQR) | Test2 Median (IQR) | ICC (95% CI) | Mean Bias (RLOA) | SEM | MDC | Cohens |
|---|---|---|---|---|---|---|---|
| Gastrocnemius Medialis (n = 87) | | | | | | | |
| Inter-rater-Test-retest | | | | | | | |
| *Fascicle Length* | | | | | | | |
| Left | 55.7 mm (49.5–62.3) | 54.0 mm (49.5–61.5) | 0.88 (0.81–0.92) | 1.5% (-21.7–31.6) | 4.4 | 12.1 | 0.11 |
| Right | 53.7 mm (49.3–63.6) | 54.6 mm (49.6–62.6) | 0.92 (0.87–0.94) | 0.1% (-18.9–23.5) | 3.5 | 9.8 | 0.03 |
| *Pennation Angle* | | | | | | | |
| Left | 18.7° (15.8–21.5) | 19.4° (16.8–22.2) | 0.75 (0.62–0.84) | -3.4% (-34.4–42.1) | 2.1 | 5.9 | 0.15 |
| Right | 19.0° (15.8–20.7) | 19.1° (16.8–21.1) | 0.64 (0.45–0.77) | -3.4% (-36.2–46.2) | 2.3 | 6.3 | 0.14 |
| *Thickness* | | | | | | | |
| Left | 15.9 mm (14.0–18.1) | 16.5 mm (14.9–18.7) | 0.86 (0.78–0.91) | -2.7% (-24.0–24.6) | 1.1 | 3.2 | 0.22 |
| Right | 16.3 mm (14.5–18.3) | 16.8 mm (14.9–18.6) | 0.83 (0.74–0.89) | -3.4% (-25.6–25.5) | 1.2 | 3.2 | 0.27 |
| Intra-rater-Test-retest | | | | | | | |
| *Fascicle Length* | | | | | | | |
| Left | 55.4 mm (48.8–62.3) | 54.2 mm (48.8–62.0) | 0.88 (0.82–0.92) | 1.2% (-21.8–30.9) | 4.3 | 12.0 | 0.09 |
| Right | 53.6 mm (48.5–62.7) | 54.1 mm (48.8–61.9) | 0.91 (0.86–0.94) | -0.1% (-20.1–24.9) | 3.7 | 10.3 | 0.01 |
| *Pennation Angle* | | | | | | | |
| Left | 18.7° (16.3–21.9) | 19.5° (17.0–22.3) | 0.76 (0.63–0.84) | -2.9% (-33.2–41.3) | 2.1 | 5.7 | 0.13 |
| Right | 19.2° (16.3–20.9) | 19.1° (17.2–21.3) | 0.63 (0.43–0.76) | -3.2% (-36.1–46.6) | 2.3 | 6.4 | 0.13 |
| *Thickness* | | | | | | | |
| Left | 15.9 mm (14.1–18.2) | 16.5 mm (14.9–18.9) | 0.86 (0.78–0.91) | -2.8% (-24.1–24.6) | 1.1 | 3.2 | 0.23 |
| Right | 16.4 mm (14.5–18.3) | 16.9 mm (15.0–18.7) | 0.83 (0.73–0.89) | -3.6% (-25.8–25.2) | 1.2 | 3.2 | 0.29 |
| Gastrocnemius Lateralis (n = 87) | | | | | | | |
| | Test1 Median (IQR) | Test2 Median (IQR) | ICC (95% CI) | Mean Bias (RLOA) | SEM | MDC | Cohens |
| Inter-rater-Test-retest | | | | | | | |
| *Fascicle Length* | | | | | | | |
| Left | 68.4 mm (59.5–77.2) | 68.7 mm (62.4–77.3) | 0.84 (0.75–0.89) | -2.0% (-26.3–30.5) | 5.7 | 15.9 | 0.13 |
| Right | 67.1 mm (59.5–77.8) | 66.3 mm (58.0–73.4) | 0.80 (0.70–0.87) | 2.8% (-25.6–42.0) | 6.2 | 17.3 | 0.15 |
| *Pennation Angle* | | | | | | | |
| Left | 12.3° (9.8–13.6) | 11.9° (9.6–13.7) | 0.79 (0.68–0.87) | 3.6% (-21.3–56.2) | 1.4 | 3.8 | 0.16 |
| Right | 12.0° (10.9–14.0) | 12.3° (10.5–13.9) | 0.61 (0.41–0.75) | -1.7% (-37.1–53.8) | 1.5 | 4.2 | 0.09 |
| *Thickness* | | | | | | | |
| Left | 12.8 mm (11.0–14.3) | 12.8 mm (10.4–14.5) | 0.88 (0.82–0.92) | 1.3% (-22.9–33.2) | 0.9 | 2.6 | 0.06 |
| Right | 13.2 mm (11.4–15.0) | 12.8 mm (10.9–14.5) | 0.81 (0.71–0.88) | 3.2% (-23.0–38.5) | 1.6 | 3.1 | 0.20 |
| Intra-rater-Test-retest | | | | | | | |
| *Fascicle Length* | | | | | | | |
| Left | 67.3 mm (59.8–76.9) | 69.3 mm (61.4–76.5) | 0.82 (0.73–0.89) | -1.9% (-27.4–32.5) | 5.9 | 16.5 | 0.13 |
| Right | 67.1 mm (59.2–78.8) | 65.8 mm (57.2–74.3) | 0.80 (0.69–0.87) | 2.0% (-27.4–43.3) | 6.5 | 18.0 | 0.08 |
| *Pennation Angle* | | | | | | | |
| Left | 12.4° (10.0–13.8) | 12.1° (9.8–14.1) | 0.79 (0.67–0.86) | 3.3% (-32.3–57.4) | 1.4 | 3.9 | 0.14 |
| Right | 12.4° (10.9–14.2) | 12.5° (10.8–14.1) | 0.63 (0.43–0.76) | -0.9% (-36.4–54.4) | 1.5 | 4.1 | 0.06 |
| *Thickness* | | | | | | | |
| Left | 12.8 mm (11.0–14.3) | 12.8 mm (10.5–14.8) | 0.88 (0.81–0.92) | 1.2% (-23.1–33.2) | 0.9 | 2.6 | 0.05 |
| Right | 13.2 mm (11.6–15.0) | 12.9 mm (11.0–14.5) | 0.81 (0.71–0.88) | 3.3% (-22.9–38.3) | 1.6 | 3.1 | 0.21 |

n = participant number; Test1 = first test session; IQR = interquartile range; Test2 = second test session; ICC = intra-class correlation coefficient; CI = confidence interval; RLOA = 95% ratio limits of agreement; SEM = standard error of measurement; MDC = minimal detectable change; mm = millimetres; ° = degrees.

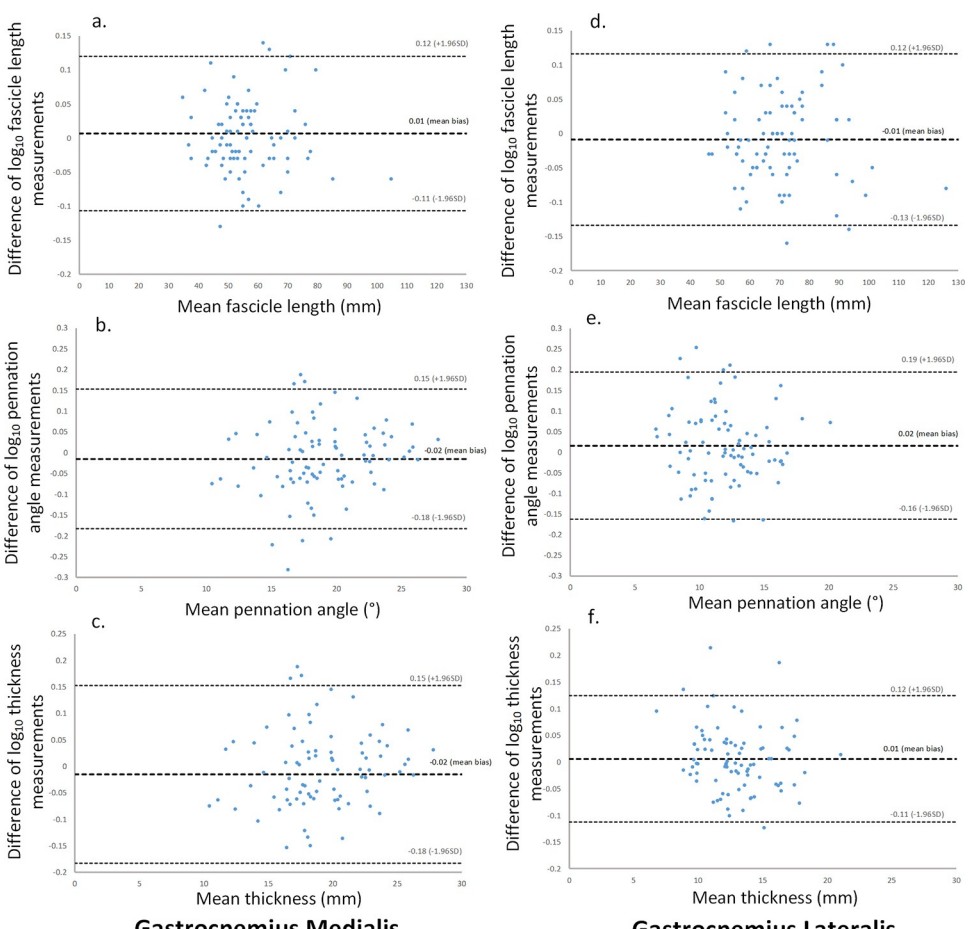

**Gastrocnemius Medialis**　　　　　　**Gastrocnemius Lateralis**

**Fig 5. Inter-rater-test-retest Bland-Altman plots of the differences of log-transformed measurements between test session 1 and test session 2 in relation to the mean of test session 1 and test session 2 measurements.** Dashed lines indicate the mean bias and the upper and lower 95% limits of agreement. Fascicle length, pennation angle, and muscle thickness measurement comparisons for the gastrocnemius medialis are represented by a. b. and c. respectively, and the gastrocnemius lateralis represented by d. e. and f.

a second occasion. Therefore, this method for test-retest analysis is unlikely to accurately track changes to an individual's gastrocnemius muscle architecture in interventional studies.

Manual digitisation is a simple process that is repeatable with excellent reliability within a single investigator and between investigators, which suggests that subjectively choosing different fascicles within a single image does not reduce reliability. Selecting three fascicles and using the average of these further improves reliability. Where direct comparisons could be made with previous studies that reported inter-rater reliability of manual digitisation of the GM but only selected one single fascicle, this study's ICCs of 0.95, 0.95, and 1.00 surpassed the reported ICCs of 0.93, 0.82, and 0.96 for FL, θ, and muscle thickness, respectively [32]. Selection of three fascicles is time efficient, only requiring an additional 5–10 seconds to complete. Future studies should consider this method rather than choosing one single fascicle. Furthermore, the lack of previous digitisation experience of investigators in this study demonstrates that manual digitisation is easy to learn with a set of instructions explaining a systematic approach to identifying anatomical landmarks. The digitiser does not need prior medical training, but basic computer skills are necessary. Of all the parameters, relative and absolute reliability was strongest for muscle thickness comparisons with narrowest RLOA. This

precision could be derived from the direct pinpointing of aponeurotic borders to find measurements of thickness, rather than selecting three random fascicles out of many to calculate FL and θ readings.

Specific to the GM, findings for FL and θ are similar or stronger than previous findings measuring reliability within investigators, where ICCs were between 0.87–0.99 for FL and between 0.85–0.99 for θ [8, 24, 25, 43, 49, 53, 72]. Between multiple investigators, this study's results for FL agree with previously reported ICCs > 0.91 [11, 32, 50]. The results for θ are stronger than previously reported values between 0.72 and 0.96 [50]. It should be noted that the reliability data presented here are specific to the protocol that this study designed. This is the first study that measures reliability of manual digitisation of FL, θ and muscle thickness between multiple investigators using the averages of 3 FLs and 3 θs from one image. The GL FLs were less reliable than the GM following repeated manual digitisation, although still excellent (ICCs > 0.95 for GM and > 0.91 for GL). This finding is likely to reflect the architectural structure of the GL muscle with longer FLs [7, 47], meaning that the ultrasound transducer field of view was often too narrow to view a long GL fascicle's origin on one aponeurosis and its insertion on the other. The transducer configuration of 4 cm to 6 cm is considered standard size and is often the cheapest and most accessible option, however when fascicles extend outside of the ultrasound image, estimating the length of the missing portion of fascicle can increase the random error in both FL and θ calculations. In specific muscles where fascicle lengths are longer, such as the GL or those in the thigh, this limitation can be overcome with specialised long transducers (10 cm) or extended-field-of-view techniques, however increased cost or time required to learn and reliably perform this technique poses some additional challenges to the clinician.

Specific to absolute test-retest reliability analyses of combined image acquisition and manual digitisation, this study found wide RLOA, indicating a large degree of variability in the method particularly when a participant returned on a separate occasion for a repeat scan. While there is some variability due to manual manipulation of the probe as it is replaced and repositioned on a participant within a single scanning session, random error was most significant when a participant was 'reset' between sessions. Skin markers, acetate paper, or plastic sheets have been used as retest aids in previous studies [25, 27, 73]. Nevertheless, this study preferentially used a system of using landmarks to set probe location, with the sonographer visually adapting the probe to obtain the best quality images per participant. Additional factors such as the amount of recent exercise to the calf, the participant inadvertently changing knee angle by self-shifting leg position on the examination table against the carefully placed foam wedge, or the size of foot in the splint impacting the angle of the ankle joint are all potential contributors to the variability between sessions and the resulting test-retest reliability. However, the small differences in the mean bias for all parameters, as well as small effect sizes between the compared groups indicates that there is no systematic error in the method used for image acquisition or manual digitisation. In terms of relative reliability, the test-retest findings align with past studies that measured GM FL and muscle thickness using protocols that use retest aids (ICC > 0.81) [8, 42, 46, 49, 53]. The comparative strength in relative reliability without the use of retest aids could be explained by the uniqueness in the current method in that the average of three images was used for session one and then compared to the average results from three images from session two. Measurements of θ were less consistent, with wider RLOA between sessions and test-retest reliability scores between moderate to good which is lower than previously reported for the GM [8, 42, 46, 49, 53]. As the analysis of manual digitisation within and between investigators revealed strong results across the three parameters, and the intra-rater between-image reliability for θ remained excellent, it can be pinpointed that θ measurements are more greatly affected by error from re-positioning of

participants and loss of precision of the location of the probe between sessions, rather than transducer orientation and rotation.

This study found that the magnitude of error of the measurement between sessions outweighs the small changes to muscle architecture that are typically seen when clinically relevant interventions are applied to gastrocnemius muscles in individuals. The changes that have been previously described to the GM muscles following interventions are frequently reported in percentages in past studies. Comparisons to previous studies can be drawn, because the descriptive measurements (median + IQR) obtained agree well with previous measurements of GM architecture in healthy adults using B-mode ultrasonography [2, 25, 35, 42, 72, 74–76]. Historically, changes seen within GM muscles following interventions are typically less than 20% for muscle thickness and even smaller values for fascicle length. One such study performed in a clinical setting found that changes to muscle thickness following 10 days of immobilisation were -15.69% in the GM, equating to approximately -2.8 mm of the mean thickness 17.9 mm (± 2.8 mm), although this was not statistically significant [74]. This study found that the MDC, or smallest change that can be detected beyond measurement error, was 3.2 mm for intra-rater-test-retest and inter-rater-test-retest analysis of GM muscle thickness. This corresponds to approximately 20% of the median muscle thickness width determined in this study of 16.4 mm (IQR: 14.6 mm to 18.7 mm). A further example can be made by exploring changes to the GM θ, which reportedly should be at least 6–7% [32] to be valid differences. One interventional study that concluded that after 5 weeks of bed rest, the GM θ declined by 14.3% [25]. An additional study found that disuse atrophy following injury (compared with uninjured leg; time period of injury not reported) resulted in a 12.7% decrease in FL, and a 16.4% decreased in θ [77]. A separate group of investigators measured GM fascicle length plasticity following 7 weeks of eccentric training in ten adult males, and showed that FL and θ significantly increased by average 7.6% and 6.8% respectively [21]. The MDC values determined in this study for GM θs ranged from 5.7˚ to 6.4˚ following the intra-rater-test-retest and inter-rater-test-retest analysis, equating to a range of 29.2% to 33.3% of the median angle values. MDC values for GM FLs ranged from 9.8 mm to 12.1 mm, or to 17.9% to 22.4% of the corresponding median FLs. These MDC values indicate that the magnitude of measurement error in using the method in this study most likely exceeds the potential changes in muscle architecture in an individual following clinical intervention such as immobilisation or eccentric training. Changes within these MDC values could not be interpreted as real although changes beyond would be considered valid. Clinicians should be aware of these MDC values when interpreting results obtained using similar techniques and consider implementing more rigorous test-retest procedures if the intention is to monitor test-retest changes in muscle architecture. An alternative and appropriate use of this method in clinical practice is to compare results between populations. The absence of systematic differences represented by the narrow mean bias supports the clinical value of evaluating differences in GM architecture between adolescent and adult group populations in a clinical setting [27].

Less data is available on plasticity of GL muscle architecture following intervention, and it cannot be assumed that adaptations observed for the GM represents the GL as different mechanical loads have been observed for the different components of the triceps surae muscle [72, 78]. The descriptive analysis highlights the differences between GM and GL architecture, consolidating the findings of previous studies that have shown that there exists a relationship between pennation angle and muscle thickness [15, 79, 80]. Muscles with greater thickness have larger pennation angles, such is seen in the GM compared to the narrower GL muscle, with corresponding smaller θs [79]. This study also confirmed that GL FLs are longer [7, 47]. One of few studies investigating changes specific to GL following training interventions found that 14 weeks of eccentric training resulted in no significant increase in FL with the change

reported to be 10%, from 86 mm ± 18 mm to 95 mm ± 28 mm [81]. A more recent study contrastingly found that adaptations were significant following 12 weeks of eccentric training, with FLs increasing by 8.8% [72]. Reductions in FL following lower limb immobilisation have also been observed in the lateral gastrocnemius with approximately 9% decrements in fascicle length after 23 days of lower limb suspension [60]. The test-retest analysis revealed that the MDC for GL FL ranged from 15.9 mm to 18.0 mm following the intra- and inter-rater analysis. This corresponds to a range of 23.1% to 27.3% of the average length of a fascicle. These MDC values for the GL are greater than the size of adaptations that have been reported in the literature. Therefore, the methods used in this study cannot confidently be used to observe true changes in muscle architecture in individuals following clinical intervention unless the changes were significantly larger than what has previously been reported in interventional studies. Including procedures employed between sessions to further standardise the probe location, such as including individualised limb moulds marked with anatomical landmarks, are likely to improve reproducibility but might be less pragmatic for use in clinical practice.

Additional limitations of this study should be acknowledged. A single trained sonographer obtained all the images used in the analysis, so this study was unable to assess inter-rater-test-retest reliability with multiple investigators performing both the image acquisition step as well as the manual digitisation step. The choice of 2D B-mode ultrasound to obtain data at-rest has been validated against cadaveric measurements; however, this modality does have a restricted field of view and uses mathematical equations to extrapolate lengths, as well as cannot appreciate the often curved, three-dimensional nature of a fascicle or aponeurosis. We followed the approach by previous authors to use the distance between two points of attachment to aponeuroses and to prioritise fascicles that were straight and parallel to the image plane to reduce the influence of omitting fascicle curvature on findings [82]. A simple model was designed to standardise the process and minimise error, which meant only GM and GL muscles were measured, with participants at rest in one position. The application of the reliability results to other positions, dynamic conditions, or other muscle groups is not recommended.

## Conclusion

In summary, the protocol employed in this study using 2D B-mode ultrasound in a clinical setting can be used to reliably measure muscle architecture of the GM and GL muscles, in a healthy adolescent or adult population. This study presented a range of reliability statistics and divided the protocol into stages for reliability analysis that allows wider comparison to previous and future literature. Within a single investigator, repeating manual digitisation alone is most reliable, followed by obtaining measures with image acquisition combined with manual digitisation within one session, then lastly obtaining measures with image acquisition combined with manual digitisation between two sessions. Comparisons between multiple investigators undertaking manual digitisation remains reliable both between images and between sessions. However, between session measurements have a high degree of variability both within an investigator or between investigators. Therefore, the ability of this protocol to determine true change to GM or GL architecture following a clinical intervention is unreliable on an individual level, as the large MDCs and RLOAs observed are greater in magnitude than the typical changes previously reported in gastrocnemius muscles after intervention. The reliability of measurements between sessions can be improved by using stricter protocols that standardise the probe location using retest aids. Further research is required to determine whether 2D B-mode ultrasonography is sensitive enough to track adaptations to muscle architecture in individual gastrocnemius muscles in a clinical setting. However, the current protocol is useful for comparisons of two population groups and can be used for this purpose in future research.

## Supporting information

**S1 Table. Inter-rater reliability: Pairwise comparisons between Investigators A, B and C for fascicle length, pennation angle and muscle thickness of the gastrocnemius medialis and gastrocnemius lateralis muscles.** IQR = interquartile range; ICC = intra-class correlation coefficient; CI = confidence interval; RLOA = ratio limits of agreement; mm = millimetres; ˚ = degrees.
(DOCX)

## Acknowledgments

The authors would like to acknowledge the contribution of Bonnie Pearson, Jack Lawrence, Nikita O'Brien, Patrick Blood, Jason Canterford and Emily Densley-Walker, who assisted with the collection and analysis of ultrasound images.

## Author Contributions

**Conceptualization:** Samantha May, Simon Locke, Michael Kingsley.

**Data curation:** Samantha May, Michael Kingsley.

**Formal analysis:** Samantha May, Michael Kingsley.

**Investigation:** Samantha May.

**Methodology:** Samantha May, Simon Locke, Michael Kingsley.

**Resources:** Michael Kingsley.

**Software:** Michael Kingsley.

**Supervision:** Simon Locke, Michael Kingsley.

**Validation:** Samantha May.

**Writing – original draft:** Samantha May.

**Writing – review & editing:** Samantha May, Simon Locke, Michael Kingsley.

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
