## [Decision Letter · Decision Letter 0]

15 Jul 2021

PONE-D-21-19762

Reliability of ultrasonographic measurement of muscle architecture of the gastrocnemius medialis and gastrocnemius lateralis

PLOS ONE

Dear Dr. Kingsley,

Thank you for submitting your manuscript to PLOS ONE. After careful consideration, we feel that it has merit but does not fully meet PLOS ONE’s publication criteria as it currently stands. Therefore, we invite you to submit a revised version of the manuscript that addresses the points raised during the review process.

ACADEMIC EDITOR:

Dear Authors, two experts in the filed revised your ms and found some major issues that you should replied in the revised ms version. In particular, consider revise the English grammar as several typos have been retrieved.

We look forward to receiving your revised manuscript.

Kind regards,

Emiliano Cè

Academic Editor

PLOS ONE

Journal Requirements:

3. We note that Figures 1 and 2 in your submission contain copyrighted images. All PLOS content is published under the Creative Commons Attribution License (CC BY 4.0), which means that the manuscript, images, and Supporting Information files will be freely available online, and any third party is permitted to access, download, copy, distribute, and use these materials in any way, even commercially, with proper attribution. For more information, see our copyright guidelines: http://journals.plos.org/plosone/s/licenses-and-copyright.

a. You may seek permission from the original copyright holder of Figures 1 and 2 to publish the content specifically under the CC BY 4.0 license. 

Reviewers' comments:

Reviewer's Responses to Questions

**Comments to the Author**

1. Is the manuscript technically sound, and do the data support the conclusions?

Reviewer #1: Yes

Reviewer #2: Yes

2. Has the statistical analysis been performed appropriately and rigorously? 

Reviewer #1: Yes

Reviewer #2: Yes

3. Have the authors made all data underlying the findings in their manuscript fully available?

Reviewer #1: Yes

Reviewer #2: Yes

4. Is the manuscript presented in an intelligible fashion and written in standard English?

Reviewer #1: Yes

Reviewer #2: Yes

5. Review Comments to the Author

Reviewer #1: This study investigates the reliability of US measurement of muscle architecture, and I assume that it is not suitable for publication for PLoS ONE after some revisions.

Please see the below comments.

Major Comments:

#1: The novelty of this study is unclear, and please emphasize this point in the introduction section.

#2: In lines 84-86, you stated the high reliability in laboratory-based protocols, but it is unclear the difference between laboratory base protocol and the clinical base protocol adopted in this study. Would you please state this point?

#3: To clarify the high reliability of US measurement, accurate image acquisition and reproducibility of analysis are required. Is there any need to consider the reproducibility of multiple analyses of a single image?

#4: Why don’t you measure the soleus muscle?

#5: In this study, the participants might be younger than the participants in clinical settings.

#6: In lines 142-143, did you measure the knee angle?

#7: The quality of Figure 3 is low, and please revise it.

#8: In this study, you measured the three fascicles in a single image, but does the reproducibility increase as the number of measured muscle fibers? Would you please add to the discussion?

#9: Why is MDC in this study greater than previous studies (lines 423-426), but the reliability of the measurement is high.

#10: I recommend that you would add some points for adapting this result in clinical practice?

Reviewer #2: This study aimed to investigate the intra-rater, inter-rater, and test-retest reliability for fascicle length, pennation angle, and muscle thickness in the frequently investigated gastrocnemii muscles. The study reported a large minimum detectable change in the measures. These findings are very important for further intervention studies.

My comments to the manuscript:

Line 30: Six images at the same spot at GM and GL? Please clarify!

Line 31: Were the three investigators also trained? Please clarify!

Line 32: I am no native speaker but before “and test-retest” should be a “,”. Read this typo several times throughout the manuscript.

Line 69: You should add here that either a panoramic mode (i.e. extended field of view mode) or special probed (i.e. 10 cm) can overcome this limitation.

Line 111: Since you mentioned it in the introduction. Would you expect differences between the reliability in GM and GL?

Line 121: Please mention the reason why there was only a single session in those subjects (E.g. dropout)

Line 143: 20-30° sounds vague. How could you confirm that you have placed the subjects similar on two different days? I know from my own experiments that varying (knee) joint angles can affect such measures in the gastrocnemii.

Line 154: typo

Line 162: Was e.g. the fascicle measured in any way standardized between the investigators? Or did the investigator pick its own chosen fascicle?

Line 182: What was the experience level of the investigators? This is very crucial

Line 269: I can imagine that the placement of the subjects (especially the pillow) might had an impact on the knee angle and hence, had an impact on the test-retest reliability. This needs to be addressed in the discussion section.

Line 327: Again, it should be mentioned here that this limitation can be addressed!

Line 400: I am not sure if this reflects the truth. Since the authors 1.) avoided to use skin markers or any other additional help 2.) the position of the subjects might vary between the test and retest (i.e. knee angle due to the pillow) I guess the minimum detectable change was likely overestimated in this study (compared to intervention studies with a more standardized setup)! Moreover, several interventions studies used larger US probes or used the panoramic mode, which likely has a better reproducibility.

Discussion in general: The authors have to discuss how the reliability can be improved in further studies! E.g. with skin markers and/or B-mode images (at a further screen) from former measures which would allow a better reproduction!

6. PLOS authors have the option to publish the peer review history of their article (what does this mean?). If published, this will include your full peer review and any attached files.

Reviewer #1: No

Reviewer #2: No

---

## [Author Response · Author response to Decision Letter 0]

25 Aug 2021

Dear Reviewers and Academic Editor,

We thank both expert reviewers and the academic editor for reviewing this manuscript. We believe that the revisions that we have made in response to your comments have improved the manuscript. Your contributions are greatly appreciated.

Our point-by-point responses to your comments and actions are below. In addition, our edits in the revised manuscript can be identified by the red text.

Academic Editor

Comment AE1: Dear Authors, two experts in the filed revised your ms and found some major issues that you should replied in the revised ms version. In particular, consider revise the English grammar as several typos have been retrieved.

Response AE1: Thank you for the opportunity to revise our manuscript and correct typos. We have re-read the document and have used Microsoft Word’s “Editor” to assist with the correction of grammatical errors. We have improved language out the manuscript to make sentences either more clear or more concise. 

Action AE1: We have made the following edits to the manuscript:

At line 31-35, we have amended the manuscript to now read: “Three independent investigators received training, then digitised the images to determine the intra-rater, inter-rater, and test-retest reliability for fascicle length (FL), pennation angle (θ) and muscle thickness. Median FL, θ, and muscle thickness for GM and GL were 53.6-55.7 mm and 65.8-69.3 mm, 18.7-19.5° and 11.9-12.5°, and 12.8-13.2 mm and 15.9-16.9 mm, respectively.”

At line 59 to 63, we have edited the manuscript, to now read: Several limitations should be considered when using ultrasound to measure skeletal muscle architecture. Ultrasound is operator-dependent, and potential sources of error stem from probe placement or location, probe pressure, and probe orientation [32, 33]. Freehand probe placement allows for manual adjustments of the probe parallel to the fascicle plane to obtain the clearest possible image, although this ‘field-based’ approach is subject to operator bias and can reduce reliability of repeated measurements on the same subject [2, 32-38].”

At line 95 to 98, we have updated the language in the manuscript to now read: “Because of the GM’s locomotor significance and ease of identification of muscle architecture under ultrasound, it has been frequently targeted in interventional studies aimed at improving biomechanical function, as well as in clinical studies of muscle disorders [46], and immobilisation studies exploring bed rest [25] or exposure to microgravity [47].”

Lines 107 to 109 now read: “Authors analysed the reliability of image acquisition, image analysis, or both of these steps with repeated measurements, however these were performed either within a single session, across multiple sessions, or across multiple investigators [11].”

Lines 113 to 115 now read: “However, when investigating responses to interventions, it is more valuable to determine whether the method of both image acquisition and digitisation repeated on multiple occasions is reliable and sensitive enough to track adaptations in the GM.” 

Lines 120 to 124 now read: “Previous literature does not address whether the magnitude of measurement error of ultrasonography exceeds the potential changes in FL, θ, and muscle thickness following clinical interventions such as immobilisation, chronic exercise training, or disuse. The reliability of imaging the GL is far less known [11], and it cannot be assumed that the method of ultrasound imaging for GM will prove reliable for GL in the same individual or group [7, 51].”

Lines 130 to 133 now read: “The primary purpose of this study was twofold: (1) to determine the reliability of ultrasound image acquisition and manual digitisation of GM and GL muscle architecture parameters in resting conditions in a clinical setting, and (2) estimate the magnitude of difference that would be required to detect real change between or within individuals and groups.”

Lines 141 to 143 now read: “Fifty-six adults and 31 adolescents aged between 13 and 63 (44 males and 43 females: mean ± SD, 22 ± 9 years, 175 ± 12 cm, and 73 ± 16 kg) provided signed consent and attended two data collection sessions at least 7 days apart.”

Lines 299 to 201 now read: “Where the fascicles extended beyond the captured image, the software extrapolated to the aponeuroses and estimated FL, like that described in model V by Ando et al. [59].”

At line 229 to 230, we have added a comma between clauses, to now read “The majority of data breached the assumption of normality (p < 0.05), and group data are presented as median and inter-quartile range (IQR).”

At lines 251 to 254, we have added a space between the term z score, to now read “The SEM was used to calculate the MDC with the equation: MDC=SEM × √2×z score(95% CI), where the z score equals 1.96, and the multiplier of √2 is to account for the uncertainty introduced by using difference scores from measurements at 2 points in time [67, 68].”

At line 306, we have changed a comma to a semicolon and removed the word however, as both clauses can stand alone, to now read “Similar findings were seen for GL (Table 2); FL was less reliable (ICC range: 0.80 to 0.84).”

Lines 335 to 338 have been updated to now read: “Reliability is reduced when ultrasound image acquisition is repeated within a session which required the probe to be removed and replaced between scans. Reliability is further reduced when participants attend re-scanning on a second occasion. Therefore, this method for test-retest analysis is unlikely to accurately track changes to an individual’s gastrocnemius muscle architecture in interventional studies.” 

At lines 360 to 361, we found an inappropriate second space between symbols, and removed it to now read “This is the first study that measures reliability of manual digitisation of FL, θ and muscle thickness between multiple investigators using the averages of 3 FLs and 3 θs from one image.”

At lines 378 to 380, we decided that after an introductory word a comma is best, and we have added a comma after the word “Nevertheless” to now read: “Nevertheless, this study preferentially used a system of using landmarks to set probe location, with the sonographer visually adapting the probe to obtain the best quality images per participant.”

Lines 386 to 390 now read: “In terms of relative reliability, the test-retest findings align with past studies that measured GM FL and muscle thickness using protocols that use retest aids (ICC > 0.81) [8, 42, 46, 49, 53]. The comparative strength in relative reliability without the use of retest aids could be explained by the uniqueness in the current method in that the average of three images was used for session one and then compared to the average results from three images from session two.”

Lines 392 to 396 now read: “As the analysis of manual digitisation within and between investigators revealed strong results across the three parameters, and the intra-rater between-image reliability for θ remained excellent, it can be pinpointed that θ measurements are more greatly affected by error from re-positioning of participants and loss of precision of the location of the probe between sessions, rather than transducer orientation and rotation.”

Lines 417 to 422 now read: “The MDC values determined in this study for GM θs ranged from 5.7° to 6.4° following the intra-rater-test-retest and inter-rater-test-retest analysis, equating to a range of 29.2% to 33.3% of the median angle values. MDC values for GM FLs ranged from 9.8 mm to 12.1 mm, or to 17.9% to 22.4% of the corresponding median FLs. These MDC values indicate that the magnitude of measurement error in using the method in this study most likely exceeds the potential changes in muscle architecture in an individual following clinical intervention such as immobilisation or eccentric training.”

At lines 422 to 423, we removed a comma between phrases that was unnecessary, and changed the word “but” to “although” to now read: “Changes within these values could not be interpreted as real although changes beyond would be considered valid.”

Lines 433 to 435 now read: “The descriptive analysis highlights the differences between GM and GL architecture, consolidating the findings of previous studies that have shown that there exists a relationship between pennation angle and muscle thickness [15, 79, 80].”

Lines 443 to 448 now read: “The test-retest analysis revealed that the MDC for GL FL ranged from 15.9 mm to 18.0 mm following the intra- and inter-rater analysis. This corresponds to a range of 23.1% to 27.3% of the average length of a fascicle. These MDC values for the GL are greater than the size of adaptations that have been reported in the literature. Therefore, the methods used in this study cannot confidently be used to observe true changes in muscle architecture in individuals following clinical intervention unless the changes were significantly larger than what has previously been reported in interventional studies.”

At lines 481 to 482, we noticed an unnecessary comma after the word “groups”. We have removed this to now read “However, the current protocol is useful for comparisons of two population groups and can be used for this purpose in future research.”

Comment AE2.1: Please ensure that your manuscript meets PLOS ONE's style requirements, including those for file naming. The PLOS ONE style templates can be found at 

Response AE2.2: Thank you for providing your comments on style requirements and the style templates. After careful review, we acknowledge that we have missed some of the formatting requirements. We apologise for these mistakes and believe we have amended these to meet the journal style requirements. We believe we have correctly named the supporting information files. 

Action AE2.2: We have made the following changes to the manuscript:

We have used a standard font (Times New Roman) and font size (size 11) throughout the manuscript (except for Headings), and ensured text is double spaced. 

On the title page, we have centred the title at line 4 and changed it from bold case to regular text as per the formatting on the sample page. We have removed the term “Authors” prior to listing the author names. 

At line 15 to 16: we updated the current address to include the department and it now reads “#aCurrent Address: Department of Sports Medicine, Alphington Sports Medicine Clinic, Northcote, Victoria, Australia”

In the body of the manuscript, the headings have been updated to sentence case with only the first word of the heading capitalized. 

At line 140:

“Study design and participants”

At line 227:

“Statistical analysis”

At lines 261 to 262:

“Analysis of manual digitisation”

“Intra-rater reliability”

At line 268:

“Inter-rater reliability”

At lines 294 to 295: 

“Analysis of image acquisition and manual digitisation”

“Intra-rater between image reliability”

At line 301:

“Test-retest reliability”

At line 738

“Supporting information”

At line 465, we noticed that the heading “Conclusion” had not been designated as a Level 1 heading, nor was size 18 font. We have updated this to meet the journal requirements for formatting. 

Comment AE2.2: We note that you have stated that you will provide repository information for your data at acceptance. Should your manuscript be accepted for publication, we will hold it until you provide the relevant accession numbers or DOIs necessary to access your data. If you wish to make changes to your Data Availability statement, please describe these changes in your cover letter and we will update your Data Availability statement to reflect the information you provide.

Response AE2.2: We appreciate your comment and have shared our data set which can be used to replicate our study’s findings, in a public repository.

May, Samantha; Kingsley, Michael (2021): Reliability of ultrasonographic measurement of muscle architecture of the gastrocnemius medialis and gastrocnemius lateralis - Datasets. La Trobe. Dataset. https://doi.org/10.26181/611a2abef0c7e

Action AE2.2: We have uploaded the data to public repository. We have changed the statement regarding data availability and added the DOI in the cover letter. We ask if you could please allow us to update the Data Availability statement on the submission system to reflect this information. 

Comment AE2.3: We note that Figures 1 and 2 in your submission contain copyrighted images. All PLOS content is published under the Creative Commons Attribution License (CC BY 4.0), which means that the manuscript, images, and Supporting Information files will be freely available online, and any third party is permitted to access, download, copy, distribute, and use these materials in any way, even commercially, with proper attribution. For more information, see our copyright guidelines: http://journals.plos.org/plosone/s/licenses-and-copyright.

a. You may seek permission from the original copyright holder of Figures 1 and 2 to publish the content specifically under the CC BY 4.0 license. 

Response AE2.3: Thank you for your comments on ownership and copyright status of Figures 1 and 2. Both photographs in these figures were taken and owned by the manuscript author Samantha May who is the original creator and rights holder of the content. 

Figure 1 is a photo of two subjects displaying the method for setup used. The two subjects in the photo were involved in the data collection of this research, and the sonographer is one of the authors of this study. The model on the table gave written permission for the photograph to be used in the research manuscript. 

Figure 2 is an ultrasound image directly taken from the ultrasound machine used in our study by one of the authors. The model’s privacy is maintained in the image. The model was not a participant in the study as they assisted in data collection, and as stated above, gave written permission for the photograph to be used in the research manuscript. The author edited the photo on Microsoft Visio to add points of interest on the image. 

Action AE2.3: We have attached a PDF file which contains the written consent provided by the model in the photos. The author of our manuscript owns the content in Figure 1 and Figure 2 and agrees to have the CC BY license applied to our work.

Reviewer # 1

Comment 1.1: The novelty of this study is unclear, and please emphasize this point in the introduction section.

Response 1.1: Thank you for your feedback. We appreciate that you have provided us the opportunity to make this clearer. Our study is the first to explore reliability of measuring gastrocnemius muscle architecture under ultrasound with a large population sample to increase the power of the study. Our sample size is significantly larger compared to past reliability studies exploring gastrocnemius muscle architecture, where sample size numbers rest between 8-21 participants (Aggeloussis, Giannakou, Albracht, & Arampatzis, 2010; de Boer et al., 2008; Duclay, Martin, Duclay, Cometti, & Pousson, 2009; Geremia et al., 2019; Konig, Cassel, Intziegianni, & Mayer, 2014; Kurokawa, Fukunaga, & Fukashiro, 2001; Kwah, Pinto, Diong, & Herbert, 2013; Maganaris, Baltzopoulos, & Sargeant, 1998; Mohagheghi et al., 2007; Muramatsu, Muraoka, Kawakami, Shibayama, & Fukunaga, 2002; Narici et al., 1996; Raj, Bird, & Shield, 2012).

Our study is the first to measure intra-rater, inter-rater, intra-rater-between-image, intra-rater-test-retest, and inter-rater-test-retest reliability of both the gastrocnemius medialis and gastrocnemius lateralis. Previous literature on the gastrocnemius lateralis is scarce. Our study is the first to determine minimum detectable change (MDC) of test-retest analysis for gastrocnemius medialis and lateralis fascicle length, pennation angle and muscle thickness, which has never been performed previously. 

Our study explores a novel method of studying gastrocnemius medialis and lateralis muscle architecture in a resting state, as we have designed the method to be easily reproducible in a clinical setting. We specifically chose to use the conventional B-mode ultrasound with a standard transducer size (4 – 6 cm), as this is most accessible and affordable to clinicians, even though there exist more modern, advanced techniques such as MRI or extended-field-of-view ultrasound, which individually carry their own benefits and limitations. We used a portable ultrasound machine and portable examination table to reflect the sports medicine environment. Unlike previous studies, we elected not to use specific re-test equipment such as plastic sheets to mark the subject. Our method is easily reproducible in a clinical setting, and we believe that the reliability analysis of this method is of use to the everyday clinician in the field of sports medicine. 

Action 1.1: The introduction of the manuscript has been amended. 

Lines 66 to 90 now read: “Standard configurations which are most cost-effective, include a transducer sized between 4 to 6 cm. These transducer field of view dimensions are often too narrow to simultaneously view both a fascicle’s origin on one aponeurosis and its insertion on the other. Specialised long transducers (10 cm) can be employed; however these transducers are more costly and increase susceptibility to uneven probe pressure spread across the underlying tissues that can cause deformation [39, 40]. More advanced extended-field-of-view techniques, where entire fascicles are imaged by moving the probe with a continuous scan, increase the risk of technical error during digital reconstruction of the image [41]. In cases where the above techniques are not available and fascicles extend beyond the ultrasound’s field of view, the external segment can be estimated by linear extrapolation of the fascicle to its intersection with the aponeuroses [2, 24, 25, 42]. Linear extrapolation assumes a fascicle follows a linear path and does not account for fascicle or aponeurosis curvature, which has important implications for the accurate calculation of architecture [43, 44]. Consequently, the reproducibility of conventional B-mode ultrasound measures of skeletal muscle architecture requires investigation. 

Authors of a systematic review found that with appropriate operator training, ultrasound measures of muscle architecture are highly reliable and valid [11]. However, most of the studies in the review described strict protocols to reduce operator bias and error, used expensive machines, or used equipment that cannot be transported to other relevant sites where sports medicine services are engaged, such as athletic clubrooms [45]. The procedures that standardise positioning of the subject and ultrasound transducer, scan parameters, image processing and digitisation analysis are time consuming, which poses a challenge to the suitability of the use of ultrasound in a clinical setting where time and equipment are often limited [40]. Therefore, the reproducibility of findings in a controlled environment might not translate to a clinical setting, where high resolution stationary ultrasound machines or retest aids are not universally available. Exploration of the reproducibility of an ultrasound method applicable to a clinical sports medicine context has not been performed. Regardless of this, conventional B-mode ultrasound imaging remains the most widely used modality to measure skeletal muscle architecture in clinical settings, due to its practicality and affordability [39].”

Lines 120 to 127 now read: “Previous literature does not address whether the magnitude of measurement error of ultrasonography exceeds the potential changes in FL, θ, and muscle thickness following clinical interventions such as immobilisation, chronic exercise training, or disuse. The reliability of imaging the GL is far less known [11], and it cannot be assumed that the method of ultrasound imaging for GM will prove reliable for GL in the same individual or group [7, 51]. Separate analysis is required, particularly test-retest reliability estimates including the minimum detectable change. This new knowledge can inform the appropriateness of tracking architectural adaptations to both heads of the gastrocnemius muscle in individuals using ultrasound.”

Comment 1.2: In lines 84-86, you stated the high reliability in laboratory-based protocols, but it is unclear the difference between laboratory base protocol and the clinical base protocol adopted in this study. Would you please state this point?

Response 1.2: Thank you for highlighting this issue. We agree that this has not been well explained and is confusing. We have considered this carefully and have decided to remove the term “laboratory-based”. This term was used to describe the nature of the strict protocols that have been applied in previous studies that explore the reliability of ultrasound measures of muscle architecture, however, is incorrect, as some studies specify that these strict protocols were performed in a laboratory environment, however some studies omit the location and hence this term might not apply. Regardless of the location of the testing, we found that past papers protocols were very time consuming and often required specific or expensive equipment that is not readily available in sports medicine, physiotherapeutic or general practice clinics. One example of this, is the use of stationary or high-precision machines (Reeves, Narici, & Maganaris, 2004), which do improve resolution of images, but are bulky, sit on heavy platforms, and cannot be easily moved off-site. We opted to use a portable ultrasound machine with a standard 38mm linear commercial transducer. The ultrasound machine chosen is low cost and very small and lightweight which is more attractive for clinical use, particularly in the field of sports medicine where consultations are often performed in athletic clubrooms. To further reduce costs and improve accessibility, we opted to use the cheapest probe, which was a 38mm linear transducer, rather than ordering a specialised wider transducer which is more expensive. To improve repeatability of our method in the sports medicine community, we chose to use static imaging rather than extended field of view imaging, which takes at least 2 weeks of dedicated training for novice sonographers to obtain high quality images (Adkins, Franks, & Murray, 2017), whereas our sonographer was able to take high quality static images after one hour of training. To simplify the process of patient setup, we used equipment that is readily available and cheap to purchase, such as an ankle night splint (which we slightly altered to remove material padding to ensure hygienic control between participants), and a bed wedge pillow with an angle of 20° to support knee flexion, which improves repeatability of the method by removing the effect of ankle dorsiflexion tensioning the gastrocnemii. 

To simplify the steps to obtain probe position, we measured the length of the tibia to find 30% distal to the knee crease as performed in previous studies, however, did not measure the exact width using tape, as this can be performed well and more efficiently by brief inspection and palpation. We designed the method so that position of the probe and depth of scanning could be manually adjusted around the site on the gastrocnemius to obtain the clearest image possible, with the probe parallel to the orientation of the fascicles and selecting an image where curvature of fascicles and aponeurosis was as reduced as possible. Other studies have not allowed for manual adjustment of the probe, rather used a method where images must be obtained exactly where the probe lies following measurement, however the quality of images and reliability of the method is then at risk if the probe is not aligned parallel to the orientation of fascicles, or if there is a high degree of curvature of the fascicle or the aponeuroses (Muramatsu et al., 2002; Narici et al., 1996). Our re-test methods were very simple compared to previous literature, where probe position has been carefully measured in terms of vertical height and horizontal width, and anatomical reference points, skin marks, and the ultrasonography scanning sites are mapped. We did not use specialised equipment such as probe straps to lock in transducer position, or plastic sheeting which has previously been used as a re-test method. Instead, we repeated the patient and probe setup at the second occasion, as the reliability for this method is much more relevant to clinical practice, where drawing marks on patients, or using re-test aids such as plastic sheeting or ultrasound probe straps is not habitually performed. 

Action 1.2: At line 52 to 90, we have amended the manuscript introduction to now read: “B-Mode ultrasound is non-invasive, readily available in research and clinical settings, and provides superior spatial resolution of images obtained in real time. This modality provides detailed visualisations of hypoechoic fibre bundles with hyperechoic septations within the muscle, separated by hyperechoic fibroadipose tissue [3, 7, 8, 15-18]. Once a single image frame or set of images are acquired, measurements are traditionally made using manual digitisation with custom-written computer software [2, 8, 19-27] or by using automatic tracking [28-31]. 

Several limitations should be considered when using ultrasound to measure skeletal muscle architecture. Ultrasound is operator-dependent, and potential sources of error stem from probe placement or location, probe pressure, and probe orientation [32, 33]. Freehand probe placement allows for manual adjustments of the probe parallel to the fascicle plane to obtain the clearest possible image, although this ‘field-based’ approach is subject to operator bias and can reduce reliability of repeated measurements on the same subject [2, 32-38]. Misalignment of the probe plane with respect to the fascicle plane can lead to over- or under-estimation of FL and θ [32, 37]. Excessive probe pressure can lead to tissue deformation and reduce the accuracy of measurements [32]. Standard configurations which are most cost-effective, include a transducer sized between 4 to 6 cm. These transducer field of view dimensions are often too narrow to simultaneously view both a fascicle’s origin on one aponeurosis and its insertion on the other. Specialised long transducers (10 cm) can be employed; however, these transducers are more costly and increase susceptibility to uneven probe pressure spread across the underlying tissues that can cause deformation [39, 40]. More advanced extended-field-of-view techniques, where entire fascicles are imaged by moving the probe with a continuous scan, increase the risk of technical error during digital reconstruction of the image [41]. In cases where the above techniques are not available and fascicles extend beyond the ultrasound’s field of view, the external segment can be estimated by linear extrapolation of the fascicle to its intersection with the aponeuroses [2, 24, 25, 42]. Linear extrapolation assumes a fascicle follows a linear path and does not account for fascicle or aponeurosis curvature, which has important implications for the accurate calculation of architecture [43, 44]. Consequently, the reproducibility of conventional B-mode ultrasound measures of skeletal muscle architecture requires investigation. 

Authors of a systematic review found that with appropriate operator training, ultrasound measures of muscle architecture are highly reliable and valid [11]. However, most of the studies in the review described strict protocols to reduce operator bias and error, used expensive machines, or used equipment that cannot be transported to other relevant sites where sports medicine services are engaged, such as athletic clubrooms [45]. The procedures that standardise positioning of the subject and ultrasound transducer, scan parameters, image processing and digitisation analysis are time consuming, which poses a challenge to the suitability of the use of ultrasound in a clinical setting where time and equipment are often limited [40]. Therefore, the reproducibility of findings in a controlled environment might not translate to a clinical setting, where high resolution stationary ultrasound machines or retest aids are not universally available. Exploration of the reproducibility of an ultrasound method applicable to a clinical sports medicine context has not been performed. Regardless of this, conventional B-mode ultrasound imaging remains the most widely used modality to measure skeletal muscle architecture in clinical settings, due to its practicality and affordability [39].”

Line 133 to 137 now reads: “The authors hypothesised that: (a) the intra-rater and inter-rater reliability of the manual digitisation step would be excellent and align with results from previous studies, and (b) that the test-retest reproducibility of the muscle architecture results between ultrasound sessions using methods repeated in a clinical setting would be lower than previously described results in the literature that applied stricter conditions or used equipment not readily available to the sports medicine clinician.”

Comment 1.3: To clarify the high reliability of US measurement, accurate image acquisition and reproducibility of analysis are required. Is there any need to consider the reproducibility of multiple analyses of a single image?

Response 1.3: Thank you for your question on reliability analysis of a single image. We believe that the intra-rater and inter-rater analysis we have performed in this study answers the question that you have posed about reproducibility of multiple analyses of a single image. Intra-rater analysis involved a single investigator repeating manual digitisation of the same, single image on two occasions, performed on 100 images. Inter-rater analysis involved comparison of the results of manual digitisation of 2145 images between three investigators. We believe that the number of images that were manually digitised and included in the intra- and inter-rater reliability analysis (100 and 2145 respectively), improves the power of the study, and reduces the margin of error. 

Action 1.3: The section of the manuscript that explains how we determined each reliability analysis, between lines 212 to 217 reads: “Fig 3 describes the methods used to determine reliability for the steps of the procedure. Intra-rater reliability of the digitisation method was analysed using one blinded investigator (Investigator A) repeating digitisation on 100 randomly sampled images on two occasions. Three blinded investigators (A, B and C) analysed all 2145 images to provide comparisons for inter-rater reliability of the digitisation method. The images were divided into images of GM (1081 images) or GL (1064 images), and reliability analyses were performed for these groups separately.”

We have also reuploaded Fig 3 as file “Fig3.tif”, with improved quality. 

Comment 1.4: Why don’t you measure the soleus muscle?

Response 1.4: Thank you for this question. The soleus is more complex than the gastrocnemius in its architecture. It contains longer fascicles, which are orientated in different directions because the soleus is compartmentalised, with a unipennate posterior section, and a bipennate anterior section (Agur, Ng-Thow-Hing, Ball, Fiume, & McKee, 2003; Hodgson, Finni, Lai, Edgerton, & Sinha, 2006), which was described in detail by Hogdson et al. (2006). This three-dimensional architecture is less appropriate to measure using conventional B-mode two-dimensional ultrasound techniques. Conventional transducers (4 – 6 cm) cannot fit in the longer fascicles within the transducer field of view, and orientation of the transducer in the plane of fascicles and perpendicular to the posterior or anterior aponeurosis is much more difficult due to curvature (Bolsterlee, Gandevia, & Herbert, 2016; Rana, Hamarneh, & Wakeling, 2014). The soleus is also deeper to the gastrocnemius, so image quality can be poor. 

Action 1.4: We have taken no specific action in amending the manuscript in response to this comment.

Comment 1.5: In this study, the participants might be younger than the participants in clinical settings.

Response 1.5: Thank you for this comment. We recruited 56 adults and 31 adolescents for this study. The age range is between 13 and 63 (44 males and 43 females: mean ± SD, 22 ± 9 years). We did not limit the upper age of our participants. Participants were excluded if they were aged 12 or younger, had a history of an acute or chronic lower limb injury within the previous 12 months, or if they had exercised their calf muscles earlier on the day of the assessment. All participants were recruited either in a sports medicine clinic, or in the rooms of local athletic clubs. According to the findings of a paper by Finch and Kenihan (2001) (Finch & Kenihan, 2001), the mean age in our study does reflect the profile of patients who attend sports medicine clinics (median age 25.4 years, range 6.8-81.6).

Action 1.5: We have taken no further action in amending the manuscript.

Comment 1.6: In lines 142-143, did you measure the knee angle?

Response 1.6: Thank you for this question. We apologise for not making this clear. We did test measurements of knee angle with a manual goniometer for the first 20 subjects, to ensure the design of our foam wedge and selected position of the wedge underneath the knee joint lines maintained a knee angle of at least 20° and limited to 30°. The participants leg would rest directly on the splint, with the tibia parallel to the 20° angle the wedge created. Knee joint angles can affect the standardisation of repeat measures. We specifically chose the angle knee flexion was to be at minimum 20° flexion at rest. The gastrocnemius bridges both the knee and ankle joint, and is under full tension when the knee is extended, because the muscle origin is furthest from its insertion (Baumbach et al., 2014). Knee flexion of at least 20° eliminates the effect of ankle dorsiflexion restraining or tensioning the gastrocnemius (Baumbach et al., 2014). 

In terms of the upper limit being chosen at 30°, this is because Baumbach et. al (2014) found that no significant differences to ankle dorsiflexion and hence gastrocnemius tensioning occurring when comparing 20° degrees of knee flexion to 30°, 45°, 60° and 75° degrees of flexion (Baumbach et al., 2014). However, we decided that 20° to 75° was too wide a range which risked increasing variability or error between measures. To standardise the measure, we limited the upper angle to 30°, as this 10° of variability is thought not to affect gastrocnemius tensioning. 

Action 1.6: At lines 165 to 172, we have added to the manuscript: “Participants lay prone on an examination table with the lower leg supported on an inclined foam wedge angled at 20° so that the knee was flexed to the within the desired range of 20-30° [54]. The wedge was consistently placed in the same position with the thin edge adjacent to the tibiofemoral joint lines, to ensure the retest position would be maintained; the knee angle was confirmed on the first 20 test subjects with a manual goniometer. The gastrocnemius bridges both the knee and ankle joint and is under full tension when the knee is extended because the muscle origin is furthest from its insertion [54]. Knee flexion of at least 20° eliminates the effect of ankle dorsiflexion restraining the gastrocnemius, and ankle dorsiflexion and gastrocnemius tension remains unchanged between 20° and 75° of knee flexion [54].”

Comment 1.7: The quality of Figure 3 is low, and please revise it.

Response 1.7: Thank you for your comment. We have revised Figure 3 to improve the image quality. 

Action 1.7: We have reuploaded Fig3.tif file. 

Comment 1.8: In this study, you measured the three fascicles in a single image, but does the reproducibility increase as the number of measured muscle fibers? Would you please add to the discussion?

Response 1.8: Thank you for your question on whether reproducibility increases if number of measured fascicles increase. The results for a pilot test analysis of 100 images, where a single fascicle in one image was selected, repeated on two occasions was ICC = 0.86, 95% CI: 0.80 - 0.90. The ICC values were calculated as per intra-rater analysis (ICC; two-way mixed-effects model, absolute definition, single rater type). In comparison, the method used in this study, where the average of three fascicles from one image was used in analysis, resulted in intra-rater reliability: ICC = 0.99, 95% CI: 0.98 - 0.99. 

This test analysis confirms that one single fascicle may not be representative of the potential diversity or heterogeneity of the fascicles within the remaining muscle volume (Bennett, Rider, Domire, DeVita, & Kulas, 2014). Previous studies that reported ICCs following repeat selection of a single fascicle within one image also reported lower reliability than our study’s method (Konig et al., 2014). Hence, we proceeded with the method of using the average of three fascicles selected within one image. 

Action 1.8: We have revised the discussion in the manuscript between lines 340 to 350, to now read: “Manual digitisation is a simple process that is repeatable with excellent reliability within a single investigator and between investigators, which suggests that subjectively choosing different fascicles within a single image does not reduce reliability. Selecting three fascicles and using the average of these further improves reliability. Where direct comparisons could be made with previous studies that reported inter-rater reliability of manual digitisation of the GM but only selected one single fascicle, this study’s ICCs of 0.95, 0.95, and 1.00 surpassed the reported ICCs of 0.93, 0.82, and 0.96 for FL, θ, and muscle thickness, respectively [32]. Selection of three fascicles is time efficient, only requiring an additional 5 – 10 seconds to complete. Future studies should use this method rather than choosing one single fascicle. Furthermore, the lack of previous digitisation experience of investigators in this study demonstrates that manual digitisation is easy to learn with a set of instructions explaining a systematic approach to identifying anatomical landmarks. The digitiser does not need prior medical training, but basic computer skills are necessary.”

Comment 1.9: Why is MDC in this study greater than previous studies (lines 423-426), but the reliability of the measurement is high.

Response 1.9: Thank you for your comment. We acknowledge that our wording is unclear. Previous studies have not reported minimum detectable change. Therefore, the MDC in this study cannot be compared to previous studies. However, the MDC results found following our research is greater than the reported adaptations in muscle architecture after intervention, reported in previous literature. 

Action 1.9: Lines 420-429 read: “These MDC values indicate that the magnitude of measurement error in using the method in this study most likely exceeds the potential changes in muscle architecture in an individual following clinical intervention such as immobilisation or eccentric training. Changes within these MDC values could not be interpreted as real although changes beyond would be considered valid. Clinicians should be aware of these MDC values when interpreting results obtained using similar techniques and consider implementing more rigorous test-retest procedures if the intention is to monitor test-rest changes in muscle architecture. An alternative and appropriate use of this method in clinical practice is to compare results between populations. The absence of systematic differences represented by the narrow mean bias supports the clinical value of evaluating differences in GM architecture between adolescent and adult group populations in a clinical setting [27].” 

We have also amended lines 443 to 451, which now read: “The test-retest analysis revealed that the MDC for GL FL ranged from 15.9 mm to 18.0 mm following the intra- and inter-rater analysis. This corresponds to a range of 23.1% to 27.3% of the average length of a fascicle. These MDC values for the GL are greater than the size of adaptations that have been reported in the literature. Therefore, the methods used in this study cannot confidently be used to observe true changes in muscle architecture in individuals following clinical intervention unless the changes were significantly larger than what has previously been reported in interventional studies. Including procedures employed between sessions to further standardise the probe location, such as including individualised limb moulds marked with anatomical landmarks, are likely to improve reproducibility but might be less pragmatic for use in clinical practice.”

Comment 1.10: I recommend that you would add some points for adapting this result in clinical practice?

Response 1.10: Thank you for this recommendation. We agree that this would enhance our study, which is designed to be repeatable in clinical settings. 

Action 1.10: Lines 422 to 429 in the manuscript have been amended to now read: “Changes within these MDC values could not be interpreted as real although changes beyond would be considered valid. Clinicians should be aware of these MDC values when interpreting results obtained using similar techniques and consider implementing more rigorous test-retest procedures if the intention is to monitor test-retest changes in muscle architecture. An alternative and appropriate use of this method in clinical practice is to compare results between populations. The absence of systematic differences represented by the narrow mean bias supports the clinical value of evaluating differences in GM architecture between adolescent and adult group populations in a clinical setting [27].”

Lines 445 to 451 have also been amended to read: “These MDC values for the GL are greater than the size of adaptations that have been reported in the literature. Therefore, the methods used in this study cannot confidently be used to observe true changes in muscle architecture in individuals following clinical intervention unless the changes were significantly larger than what has previously been reported in interventional studies. Including procedures employed between sessions to further standardise the probe location, such as including individualised limb moulds marked with anatomical landmarks, are likely to improve reproducibility but might be less pragmatic for use in clinical practice.”

Reviewer #2

Comment 2.0: This study aimed to investigate the intra-rater, inter-rater, and test-retest reliability for fascicle length, pennation angle, and muscle thickness in the frequently investigated gastrocnemii muscles. The study reported a large minimum detectable change in the measures. These findings are very important for further intervention studies.

Response 2.0: Thank you kindly for this positive comment about this work. We have updated our manuscript discussion to further emphasise this point which is already made in our conclusion. We also have found that the method employed in this study can be used in future studies that compare two populations.

Action 2.0: Lines 422 to 429 have been amended, and now read: “Changes within these MDC values could not be interpreted as real although changes beyond would be considered valid. Clinicians should be aware of these MDC values when interpreting results obtained using similar techniques and consider implementing more rigorous test-retest procedures if the intention is to monitor test-retest changes in muscle architecture. An alternative and appropriate use of this method in clinical practice is to compare results between populations. The absence of systematic differences represented by the narrow mean bias supports the clinical value of evaluating differences in GM architecture between adolescent and adult group populations in a clinical setting [27].”

Comment 2.1 - Line 30: Six images at the same spot at GM and GL? Please clarify!

Response 2.1: Thank you for your comment. We have updated this sentence to clarify the uncertainty in the abstract. Six images were taken in total from each participant, 3 from each of the gastrocnemius medialis, and gastrocnemius lateralis. 

Action 2.1: The manuscript has been amended between lines 30 to 31: “A trained sonographer obtained three B-mode images from each of the GM and GL muscles in 87 volunteers (44 males, 43 females; 22±9 years of age) on two separate occasions.”

Comment 2.2 - Line 31: Were the three investigators also trained? Please clarify!

Response 2.2: Thank you for your comment. The three investigators received one hour of training as they had no previous experience or training in digitisation of gastrocnemius muscle architecture. We have amended our abstract and methods section to better clarify this. 

Action 2.2: Lines 31 to 33 now read: “Three independent investigators received training, then digitised the images to determine the intra-rater, inter-rater, and test-retest reliability for fascicle length (FL), pennation angle (θ) and muscle thickness.”

Lines 173 to 176 now read: “A trained sonographer used a measuring tape to find the initial probe site at one-third of the distance from the popliteal crease of the knee to the tip of the medial malleolus for the GM and the lateral malleolus for the GL, at the mid-muscle belly which was determined via inspection and palpation [7, 27, 32, 47, 49, 53, 55].”

Lines 197 to 199 now read: “Investigators received one hour of training and were provided with step-by-step training manuals that instructed investigators to prioritise selection of fascicles that were straight with visible endpoints at the junction of either the deep or superficial aponeurosis.” 

Comment 2.3 - Line 32: I am no native speaker but before “and test-retest” should be a “,”. Read this typo several times throughout the manuscript.

Response 2.3: Thank you for your suggestion. We agree that a comma is required to separate “and test-retest” from the items in the series described prior in the same sentence. We have corrected this typo throughout the manuscript. When test-retest is further divided into inter-rater-test-retest or intra-rater-test-retest, we have decided to clarify this relationship with hyphens rather than commas to improve understanding about the reliability analysis we are describing.

Action 2.3: Lines 31 to 33 now read: “Three independent investigators received training, then digitised the images to determine the intra-rater, inter-rater, and test-retest reliability for fascicle length (FL), pennation angle (θ) and muscle thickness.”

Lines 220 to 223 now read: “Intra-rater-test-retest reliability was determined by comparing Investigator A’s averages from test session 1 images to test session 2. The average of the three investigators (A, B and C) results from test session 1 was also compared against their average of results from test session 2 to determine inter-rater-test-retest reliability.”

Lines 302 to 303 now read: “The results of the image acquisition and digitisation inter-rater-test-retest and intra-rater-test-retest analyses are presented in Table 2.”

Lines 408 to 409 now read: “This study found that the MDC, or smallest change that can be detected beyond measurement error, was 3.2 mm for intra-rater-test-retest and inter-rater-test-retest analysis of GM muscle thickness.”

Lines 417 to 419 now read: “The MDC values determined in this study for GM θs ranged from 5.7° to 6.4° following the intra-rater-test-retest and inter-rater-test-retest analysis, equating to a range of 29.2% to 33.3% of the median angle values.”

Lines 453 to 455 now read: “A single trained sonographer obtained all the images used in the analysis, so this study was unable to assess inter-rater-test-retest reliability with multiple investigators performing both the image acquisition step as well as the manual digitisation step.”

We have also updated this within Table 2 between lines 318 and 319, and the Figure 5 label at lines 323 to 325: “*** Fig 5. Inter-rater-test-retest Bland-Altman plots of the differences of log-transformed measurements between test session 1 and test session 2 in relation to the mean of test session 1 and test session 2 measurements”

Comment 2.4 Line 69: You should add here that either a panoramic mode (i.e. extended field of view mode) or special probed (i.e. 10 cm) can overcome this limitation.

Response 2.4: Thank you for this suggestion. We agree that other ultrasound techniques could overcome the limitation of narrow field of view with a commercial transducer, which are sized between 4-6 cm. We agree that this should be pointed out in this study. 

Action 2.4: Between lines 66 to 77, we have updated the manuscript introduction to now read: “Standard configurations which are most cost-effective, include a transducer sized between 4 to 6 cm. These transducer field of view dimensions are often too narrow to simultaneously view both a fascicle’s origin on one aponeurosis and its insertion on the other. Specialised long transducers (10 cm) can be employed; however, these transducers are more costly and increase susceptibility to uneven probe pressure spread across the underlying tissues that can cause deformation [39, 40]. More advanced extended-field-of-view techniques, where entire fascicles are imaged by moving the probe with a continuous scan, increase the risk of technical error during digital reconstruction of the image [41]. In cases where the above techniques are not available and fascicles extend beyond the ultrasound’s field of view, the external segment can be estimated by linear extrapolation of the fascicle to its intersection with the aponeuroses [2, 24, 25, 42]. Linear extrapolation assumes a fascicle follows a linear path and does not account for fascicle or aponeurosis curvature, which has important implications for the accurate calculation of architecture [43, 44]. Consequently, the reproducibility of conventional B-mode ultrasound measures of skeletal muscle architecture requires investigation.”

Comment 2.5 - Line 111: Since you mentioned it in the introduction. Would you expect differences between the reliability in GM and GL?

Response 2.5: Thank you for your question. The answer is yes, we did expect differences between reliability for fascicle length, and pennation angle. This is because the gastrocnemius muscles have different architectural properties; GL has longer FLs and smaller θs whereas the GM comprises shorter FLs and larger θs (Kawakami, Ichinose, & Fukunaga, 1998; Koryak, 2019). Historically, taking conventional ultrasound images with a narrow field of view of longer fascicles can reduce reliability of the manual digitisation method due to extrapolation of the fascicles using linear equations. We explored this with inter-rater analysis of 1081 GM images and 1064 GL images, with our reliability results outlined in Table 1 in the manuscript, which show excellent reliability with ICC 0.95 for GM, and 0.91 for GL fascicle length, and 0.95 for GM and 0.94 for GL pennation angle. The reduction in reliability indicates that the method we used to image GL is more prone to measurement error for FL, compared to the GM. Similar findings have been reported in past papers that explore reliability of the gastrocnemius medialis as well as a separate muscle with longer fascicles such as the vastus lateralis in the thigh (Raj et al., 2012).

Action 2.5: We have amended the discussion between lines 361 to 371 to now read: “The GL FLs were less reliable than the GM following repeated manual digitisation, although still excellent (ICCs > 0.95 for GM and > 0.91 for GL). This finding is likely to reflect the architectural structure of the GL muscle with longer FLs [7, 47], meaning that the ultrasound transducer field of view was often too narrow to view a long GL fascicle’s origin on one aponeurosis and its insertion on the other. The transducer configuration of 4 cm to 6 cm is considered standard size and is often the cheapest and most accessible option, however when fascicles extend outside of the ultrasound image, estimating the length of the missing portion of fascicle can increase the random error in both FL and θ calculations. In specific muscles where fascicle lengths are longer, such as the GL or those in the thigh, this limitation can be overcome with specialised long transducers (10 cm) or extended-field-of-view techniques, however increased cost or time required to learn and reliably perform this technique poses some additional challenges to the clinician.”

Comment 2.6 - Line 121: Please mention the reason why there was only a single session in those subjects (E.g. dropout)

Response 2.6: Thank you for your comment and we apologise for the lack of clarity. You are correct in that a single session was performed in those subjects were due to dropout.

Action 2.6: We have amended the manuscript between lines 143 to 144 to now read: “An additional 2 adult and 4 adolescent participants attended a single data collection session but did not attend the second data collection session and were not included in the test-retest reliability analysis.”

Comment 2.7 - Line 143: 20-30° sounds vague. How could you confirm that you have placed the subjects similar on two different days? I know from my own experiments that varying (knee) joint angles can affect such measures in the gastrocnemii.

Response 2.7: Thank you for your comment and expertise on this subject. We agree that knee joint angles can affect the standardisation of repeat measures. We specifically chose the angle knee flexion was to be at minimum 20° flexion at rest. The gastrocnemius bridges both the knee and ankle joint, and is under full tension when the knee is extended, because the muscle origin is furthest from its insertion (Baumbach et al., 2014). Knee flexion of at least 20° eliminates the effect of ankle dorsiflexion restraining or tensioning the gastrocnemius (Baumbach et al., 2014). To achieve this efficiently, we used a 20° foam wedge designed to sit at the angle of the knee joint. The participants leg would rest directly on the splint, with the tibia parallel to the 20° angle the wedge created, and we confirmed with test runs that the knee flexion angle was consistently >20° if the thin edge of the wedge was placed at the angle of the knee joint lines, and the tibia appeared parallel to the splint whilst resting on the splint. In terms of the upper limit being chosen at 30°, this is because Baumbach et. al (2014) found that no significant differences to ankle dorsiflexion and hence gastrocnemius tensioning occurring when comparing 20° degrees of knee flexion to 30°, 45°, 60° and 75° degrees of flexion. However, we decided that 20° to 75° was too wide a range which risked increasing variability or error between measures. To standardise the measure, we limited the upper angle to 30°, as this 10° of variability is thought not to affect gastrocnemius tensioning. We accepted any measurement between this range on our test 20 subjects, confirmed with a goniometer. 

Action 2.7: At lines 165 to 173, we have added to the manuscript: “Participants lay prone on an examination table with the lower leg supported on an inclined foam wedge angled at 20° so that the knee was flexed to the within the desired range of 20-30° [54]. The wedge was consistently placed in the same position with the thin edge adjacent to the tibiofemoral joint lines, to ensure the retest position would be maintained; the knee angle was confirmed on the first 20 test subjects with a manual goniometer. The gastrocnemius bridges both the knee and ankle joint and is under full tension when the knee is extended because the muscle origin is furthest from its insertion [54]. Knee flexion of at least 20° eliminates the effect of ankle dorsiflexion restraining the gastrocnemius, and ankle dorsiflexion and gastrocnemius tension remains unchanged between 20° and 75° of knee flexion [54]. The ankle was secured at approximately 90° using a night splint and confirmed with a manual goniometer (Fig 1).”

Comment 2.8 - Line 154: typo

Response 2.8: Thank you for finding this typo “and,”. We have removed it. 

Action 2.8: The manuscript now reads between lines 182 to 184: Multiple ultrasound images were obtained at the site where the muscle belly was widest, as this is where FLs are homogenously distributed [8, 58].”

Comment 2.9 - Line 162: Was e.g. the fascicle measured in any way standardized between the investigators? Or did the investigator pick its own chosen fascicle?

Response 2.9: Thank you for your questions, and we agree that this information is critical to providing a clear explanation of the methods of this study. Investigators received one hour of training to learn how to manually digitise the ultrasound images using bespoke software designed in LabVIEW (version 16; National Instruments, US). Training instructions included a written manual provided to the investigators, which explained stepwise how to use the program to account for the depth of the scan, and mark key points to digitise the following anatomical structures: superficial aponeurosis external and internal borders, deep aponeurosis external and internal borders, and two endpoints of three fascicles within the imaging plane. The investigators were instructed to select fascicles that were visible, straight, and with those were the endpoints could be visualised at the junction of either the deep or superficial aponeurosis. This is because the length of the fascicle is defined by the coordinates of the superficial junction and the deep junction for the fascicle of interest. If two endpoints could not be visualised, the investigators were instructed to select fascicles where one endpoint could be visualised. The software would combine the measurement of the length of the visible part of the fascicle with linear extrapolation of the length of the part of the fascicle that was not visible on the sonographic image. Therefore, within each image, investigators would subjectively pick three chosen fascicles biased towards meeting the predetermined criteria. Each investigator first digitised a test 100 images and pairwise comparisons were made, which showed there were no constant inconsistencies that would reflect one investigators error in following the provided instructions. We have amended the methods section to further describe this training and process. 

Action 2.9: The manuscript has been updated in the methods section between lines 197 to 199 to now read: “Investigators received one hour of training and were provided with step-by-step training manuals that instructed investigators to prioritise selection of fascicles that were straight with visible endpoints at the junction of either the deep or superficial aponeurosis.”

Comment 2.10 - Line 182: What was the experience level of the investigators? This is very crucial.

Response 2.10: Thank you for your question. The investigators had no prior experience in digitisation. They received one hour of training, and an instruction manual. The sonographer was a novice, who had received previous basic training in using ultrasound in clinical setting prior to the study. The sonographer received one additional hour of training to learn the methods outlined in this study to limit error and improve repeatability. This included practicing manual adjustment of the probe to align parallel with the plane of the fascicle, orientating the probe to reduce aponeuroses curvature, adjusting depth of scanning to ensure outer borders of the superficial and deep aponeuroses were visible, and taking care to reduce probe pressure.

Action 2.10: The manuscript has been updated in the methods section. Lines 173 to 176 now reads: “A trained sonographer used a measuring tape to find the initial probe site at one-third of the distance from the popliteal crease of the knee to the tip of the medial malleolus for the GM and the lateral malleolus for the GL, at the mid-muscle belly which was determined via inspection and palpation [7, 27, 32, 47, 49, 53, 55].”

Between lines 197 to 199 to now read: “Investigators received one hour of training and were provided with step-by-step training manuals that instructed investigators to prioritise selection of fascicles that were straight with visible endpoints at the junction of either the deep or superficial aponeurosis..”

The discussion section has also been updated between lines 340 to 350 to now read: “Manual digitisation is a simple process that is repeatable with excellent reliability within a single investigator and between investigators, which suggests that subjectively choosing different fascicles within a single image does not reduce reliability. Selecting three fascicles and using the average of these further improves reliability. Where direct comparisons could be made with previous studies that reported inter-rater reliability of manual digitisation of the GM but only selected one single fascicle, this study’s ICCs of 0.95, 0.95, and 1.00 surpassed the reported ICCs of 0.93, 0.82, and 0.96 for FL, θ, and muscle thickness, respectively [32]. Selection of three fascicles is time efficient, only requiring an additional 5 – 10 seconds to complete. Future studies should use this method rather than choosing one single fascicle. Furthermore, the lack of previous digitisation experience of investigators in this study demonstrates that manual digitisation is easy to learn with a set of instructions explaining a systematic approach to identifying anatomical landmarks. The digitiser does not need prior medical training, but basic computer skills are necessary.”

Comment 2.11 - Line 269: I can imagine that the placement of the subjects (especially the pillow) might had an impact on the knee angle and hence, had an impact on the test-retest reliability. This needs to be addressed in the discussion section.

Response 2.11: Thank you for your comment, and we agree that this required better explanation in the manuscript. We first clarified how we decided on the knee angle and the method we used to achieve this in the methods section. We have also acknowledged in the discussion, that patient positioning potentially contributes to variability between sessions and the resulting test-retest reliability. 

Action 2.11: In the methods section, we have updated the manuscript between lines 165 to 172 to now read: “Participants lay prone on an examination table with the lower leg supported on an inclined foam wedge angled at 20° so that the knee was flexed to the within the desired range of 20-30° [54]. The wedge was consistently placed in the same position with the thin edge adjacent to the tibiofemoral joint lines, to ensure the retest position would be maintained; the knee angle was confirmed on the first 20 test subjects with a manual goniometer. The gastrocnemius bridges both the knee and ankle joint and is under full tension when the knee is extended because the muscle origin is furthest from its insertion [54]. Knee flexion of at least 20° eliminates the effect of ankle dorsiflexion restraining the gastrocnemius, and ankle dorsiflexion and gastrocnemius tension remains unchanged between 20° and 75° of knee flexion [54].”

In the discussion section, we have updated the manuscript between lines 380 to 384 to now read: “Additional factors such as the amount of recent exercise to the calf, the participant inadvertently changing knee angle by self-shifting leg position on the examination table against the carefully placed foam wedge, and the size of foot in the splint impacting the angle of the ankle joint are all potential contributors to the variability between sessions and the resulting test-retest reliability.”

Comment 2.12 - Line 327: Again, it should be mentioned here that this limitation can be addressed!

Response 2.12: Thank you for your comment, which refers to the selected ultrasound transducer field of view intermittently being too narrow to view a long GL fascicle’s origin on one aponeurosis and its insertion on the other. We agree this limitation can be overcome with other ultrasound techniques; however, these techniques may be expensive or time-inefficient, and this must also be acknowledged as a barrier to measuring muscle architecture under ultrasound in clinical practice. 

Action 2.12: The introduction section of the manuscript has been amended between lines 66 to 77 to now read: “Standard configurations which are most cost-effective, include a transducer sized between 4 to 6 cm. These transducer field of view dimensions are often too narrow to simultaneously view both a fascicle’s origin on one aponeurosis and its insertion on the other. Specialised long transducers (10 cm) can be employed; however, these transducers are more costly and increase susceptibility to uneven probe pressure spread across the underlying tissues that can cause deformation [39, 40]. More advanced extended-field-of-view techniques, where entire fascicles are imaged by moving the probe with a continuous scan, increase the risk of technical error during digital reconstruction of the image [41]. In cases where the above techniques are not available and fascicles extend beyond the ultrasound’s field of view, the external segment can be estimated by linear extrapolation of the fascicle to its intersection with the aponeuroses [2, 24, 25, 42]. Linear extrapolation assumes a fascicle follows a linear path and does not account for fascicle or aponeurosis curvature, which has important implications for the accurate calculation of architecture [43, 44]. Consequently, the reproducibility of conventional B-mode ultrasound measures of skeletal muscle architecture requires investigation.”

The manuscript discussion section has been amended between lines 363 to 371 to now read: “This finding is likely to reflect the architectural structure of the GL muscle with longer FLs [7, 47], meaning that the ultrasound transducer field of view was often too narrow to view a long GL fascicle’s origin on one aponeurosis and its insertion on the other. The transducer configuration of 4 cm to 6 cm is considered standard size and is often the cheapest and most accessible option, however when fascicles extend outside of the ultrasound image, estimating the length of the missing portion of fascicle can increase the random error in both FL and θ calculations. In specific muscles where fascicle lengths are longer, such as the GL or those in the thigh, this limitation can be overcome with specialised long transducers (10 cm) or extended-field-of-view techniques, however increased cost or time required to learn and reliably perform this technique poses some additional challenges to the clinician.”

Comment 2.12 - Line 400: I am not sure if this reflects the truth. Since the authors 1.) avoided to use skin markers or any other additional help 2.) the position of the subjects might vary between the test and retest (i.e. knee angle due to the pillow) I guess the minimum detectable change was likely overestimated in this study (compared to intervention studies with a more standardized setup)! Moreover, several interventions studies used larger US probes or used the panoramic mode, which likely has a better reproducibility.

Response 2.12: Thank you for this comment. We specifically designed our method to be pragmatic in a clinical environment. Therefore, our MDC reflects the nature of the method we employed. We do agree that employing tighter controls could reduce the MDC, however these controls may not be representative of a clinical setting. For example, manual adaptation of the probe at the site of scanning is required to obtain best quality images, which can create variability in probe location.

In terms of comparison to other studies, we cannot comment on MDC as previous studies chose not to reported this. However, we do agree that our results reflect that random error was greatest when a participant returned on a separate occasion for a repeat scan, which we believe is due to the participant being ‘reset’ between sessions. Skin markers, acetate paper, or plastic sheets have been used as retest aids in previous studies, but this is not necessarily convenient or easily accessible in a clinical environment. 

Action 2.12: We have updated the manuscript to reflect our comments. 

Between lines 373 to 396, the manuscript has been amended to now read: “Specific to absolute test-retest reliability analyses of combined image acquisition and manual digitisation, this study found wide RLOA, indicating a large degree of variability in the method particularly when a participant returned on a separate occasion for a repeat scan. While there is some variability due to manual manipulation of the probe as it is replaced and repositioned on a participant within a single scanning session, random error was most significant when a participant was ‘reset’ between sessions. Skin markers, acetate paper, or plastic sheets have been used as retest aids in previous studies [25, 27, 73]. Nevertheless, this study preferentially used a system of using landmarks to set probe location, with the sonographer visually adapting the probe to obtain the best quality images per participant. Additional factors such as the amount of recent exercise to the calf, the participant inadvertently changing knee angle by self-shifting leg position on the examination table against the carefully placed foam wedge, and the size of foot in the splint impacting the angle of the ankle joint are all potential contributors to the variability between sessions and the resulting test-retest reliability. However, the small differences in the mean bias for all parameters, as well as small effect sizes between the compared groups indicates that there is no systematic error in the method used for image acquisition nor manual digitisation. In terms of relative reliability, the test-retest findings align with past studies that measured GM FL and muscle thickness using protocols that use retest aids (ICC > 0.81) [8, 42, 46, 49, 53]. The comparative strength in relative reliability without the use of retest aids could be explained by the uniqueness in the current method in that the average of three images was used for session one and then compared to the average results from three images from session two. Measurements of θ were less consistent, with wider RLOA between sessions and test-retest reliability scores between moderate to good which is lower than previously reported for the GM [8, 42, 46, 49, 53]. As the analysis of manual digitisation within and between investigators revealed strong results across the three parameters, and the intra-rater between-image reliability for θ remained excellent, it can be pinpointed that θ measurements are more greatly affected by error from re-positioning of participants and loss of precision of the location of the probe between sessions, rather than transducer orientation and rotation.”

In addition, lines 445 to 451 have been amended to read: “These MDC values for the GL are greater than the size of adaptations that have been reported in the literature. Therefore, the methods used in this study cannot confidently be used to observe true changes in muscle architecture in individuals following clinical intervention unless the changes were significantly larger than what has previously been reported in interventional studies. Including procedures employed between sessions to further standardise the probe location, such as including individualised limb moulds marked with anatomical landmarks, are likely to improve reproducibility but might be less pragmatic for use in clinical practice.”

Comment 2.13: Discussion in general: The authors have to discuss how the reliability can be improved in further studies! E.g. with skin markers and/or B-mode images (at a further screen) from former measures which would allow a better reproduction!

Response 2.13: Thank you for this comment. We agree that there exist more controlled methods and modern ultrasound techniques that could improve reproducibility and reliability of the method. We have added these points to the manuscript. 

Action 2.13: The manuscript has been amended at lines 363 to 371 to now read: “This transducer configuration of 4 to 6 cm is considered standard size and is often the cheapest and most accessible option, however when fascicles extend off the ultrasound image, the estimate of the length of the missing portion of fascicle must be used and may increase the random error in both FL and θ calculations. In specific muscles where fascicle lengths are longer, such as the GL or those in the thigh, this limitation can be overcome with specialised long transducers (10 cm) or extended-field-of-view techniques, however increased cost or time required to learn and reliably perform this technique poses other unique challenges to the clinician.”

Between lines 445 and 451 now read: “These MDC values for the GL are greater than the size of adaptations that have been reported in the literature. Therefore, the methods used in this study cannot confidently be used to observe true changes in muscle architecture in individuals following clinical intervention unless the changes were significantly larger than what has previously been reported in interventional studies. Including procedures employed between sessions to further standardise the probe location, such as including individualised limb moulds marked with anatomical landmarks, are likely to improve reproducibility but might be less pragmatic for use in clinical practice.”

The conclusion between lines 477 and 479 remains: “The reliability of measurements between sessions can be improved by using stricter protocols that standardise the probe location using re-test aids.”

Additional References in the Response Document

Adkins, A. N., Franks, P. W., & Murray, W. M. (2017). Demonstration of extended field-of-view ultrasound's potential to increase the pool of muscles for which in vivo fascicle length is measurable. J Biomech, 63, 179-185. doi:10.1016/j.jbiomech.2017.08.012

Aggeloussis, N., Giannakou, E., Albracht, K., & Arampatzis, A. (2010). Reproducibility of fascicle length and pennation angle of gastrocnemius medialis in human gait in vivo. Gait Posture, 31(1), 73-77. doi:10.1016/j.gaitpost.2009.08.249

Agur, A. M., Ng-Thow-Hing, V., Ball, K. A., Fiume, E., & McKee, N. H. (2003). Documentation and three-dimensional modelling of human soleus muscle architecture. Clin Anat, 16(4), 285-293. doi:10.1002/ca.10112

Baumbach, S. F., Brumann, M., Binder, J., Mutschler, W., Regauer, M., & Polzer, H. (2014). The influence of knee position on ankle dorsiflexion - a biometric study. BMC Musculoskelet Disord, 15, 246. doi:10.1186/1471-2474-15-246

Bennett, H. J., Rider, P. M., Domire, Z. J., DeVita, P., & Kulas, A. S. (2014). Heterogeneous fascicle behavior within the biceps femoris long head at different muscle activation levels. J Biomech, 47(12), 3050-3055. doi:10.1016/j.jbiomech.2014.06.032

Bolsterlee, B., Gandevia, S. C., & Herbert, R. D. (2016). Effect of Transducer Orientation on Errors in Ultrasound Image-Based Measurements of Human Medial Gastrocnemius Muscle Fascicle Length and Pennation. PLoS One, 11(6), e0157273. doi:10.1371/journal.pone.0157273

de Boer, M. D., Seynnes, O. R., di Prampero, P. E., Pisot, R., Mekjavic, I. B., Biolo, G., & Narici, M. V. (2008). Effect of 5 weeks horizontal bed rest on human muscle thickness and architecture of weight bearing and non-weight bearing muscles. Eur J Appl Physiol, 104(2), 401-407. doi:10.1007/s00421-008-0703-0

Duclay, J., Martin, A., Duclay, A., Cometti, G., & Pousson, M. (2009). Behavior of fascicles and the myotendinous junction of human medial gastrocnemius following eccentric strength training. Muscle Nerve, 39(6), 819-827. doi:10.1002/mus.21297

Finch, C. F., & Kenihan, M. A. (2001). A profile of patients attending sports medicine clinics. Br J Sports Med, 35(4), 251-256. doi:10.1136/bjsm.35.4.251

Geremia, J. M., Baroni, B. M., Bini, R. R., Lanferdini, F. J., de Lima, A. R., Herzog, W., & Vaz, M. A. (2019). Triceps Surae Muscle Architecture Adaptations to Eccentric Training. Front Physiol, 10, 1456. doi:10.3389/fphys.2019.01456

Hodgson, J. A., Finni, T., Lai, A. M., Edgerton, V. R., & Sinha, S. (2006). Influence of structure on the tissue dynamics of the human soleus muscle observed in MRI studies during isometric contractions. J Morphol, 267(5), 584-601. doi:10.1002/jmor.10421

Kawakami, Y., Ichinose, Y., & Fukunaga, T. (1998). Architectural and functional features of human triceps surae muscles during contraction. J Appl Physiol (1985), 85(2), 398-404. Retrieved from https://www.ncbi.nlm.nih.gov/pubmed/9688711

Konig, N., Cassel, M., Intziegianni, K., & Mayer, F. (2014). Inter-rater reliability and measurement error of sonographic muscle architecture assessments. J Ultrasound Med, 33(5), 769-777. doi:10.7863/ultra.33.5.769

Koryak, Y. A. (2019). Architectural and functional specifics of the human triceps surae muscle in vivo and its adaptation to microgravity. J Appl Physiol (1985), 126(4), 880-893. doi:10.1152/japplphysiol.00634.2018

Kurokawa, S., Fukunaga, T., & Fukashiro, S. (2001). Behavior of fascicles and tendinous structures of human gastrocnemius during vertical jumping. J Appl Physiol (1985), 90(4), 1349-1358. doi:10.1152/jappl.2001.90.4.1349

Kwah, L. K., Pinto, R. Z., Diong, J., & Herbert, R. D. (2013). Reliability and validity of ultrasound measurements of muscle fascicle length and pennation in humans: a systematic review. J Appl Physiol (1985), 114(6), 761-769. doi:10.1152/japplphysiol.01430.2011

Maganaris, C. N., Baltzopoulos, V., & Sargeant, A. J. (1998). In vivo measurements of the triceps surae complex architecture in man: implications for muscle function. J Physiol, 512 ( Pt 2), 603-614. doi:10.1111/j.1469-7793.1998.603be.x

Mohagheghi, A. A., Khan, T., Meadows, T. H., Giannikas, K., Baltzopoulos, V., & Maganaris, C. N. (2007). Differences in gastrocnemius muscle architecture between the paretic and non-paretic legs in children with hemiplegic cerebral palsy. Clin Biomech (Bristol, Avon), 22(6), 718-724. doi:10.1016/j.clinbiomech.2007.03.004

Muramatsu, T., Muraoka, T., Kawakami, Y., Shibayama, A., & Fukunaga, T. (2002). In vivo determination of fascicle curvature in contracting human skeletal muscles. J Appl Physiol (1985), 92(1), 129-134. doi:10.1152/jappl.2002.92.1.129

Narici, M. V., Binzoni, T., Hiltbrand, E., Fasel, J., Terrier, F., & Cerretelli, P. (1996). In vivo human gastrocnemius architecture with changing joint angle at rest and during graded isometric contraction. J Physiol, 496 ( Pt 1), 287-297. doi:10.1113/jphysiol.1996.sp021685

Raj, I. S., Bird, S. R., & Shield, A. J. (2012). Reliability of ultrasonographic measurement of the architecture of the vastus lateralis and gastrocnemius medialis muscles in older adults. Clin Physiol Funct Imaging, 32(1), 65-70. doi:10.1111/j.1475-097X.2011.01056.x

Rana, M., Hamarneh, G., & Wakeling, J. M. (2014). 3D curvature of muscle fascicles in triceps surae. J Appl Physiol (1985), 117(11), 1388-1397. doi:10.1152/japplphysiol.00109.2013

Reeves, N. D., Narici, M. V., & Maganaris, C. N. (2004). In vivo human muscle structure and function: adaptations to resistance training in old age. Exp Physiol, 89(6), 675-689. doi:10.1113/expphysiol.2004.027797

---

## [Decision Letter · Decision Letter 1]

16 Sep 2021

Reliability of ultrasonographic measurement of muscle architecture of the gastrocnemius medialis and gastrocnemius lateralis

PONE-D-21-19762R1

Dear Dr. Kingsley,

We’re pleased to inform you that your manuscript has been judged scientifically suitable for publication and will be formally accepted for publication once it meets all outstanding technical requirements.

Kind regards,

Emiliano Cè

Academic Editor

PLOS ONE

Additional Editor Comments (optional):

Reviewers' comments:

Reviewer's Responses to Questions

**Comments to the Author**

1. If the authors have adequately addressed your comments raised in a previous round of review and you feel that this manuscript is now acceptable for publication, you may indicate that here to bypass the “Comments to the Author” section, enter your conflict of interest statement in the “Confidential to Editor” section, and submit your "Accept" recommendation.

Reviewer #1: All comments have been addressed

Reviewer #2: All comments have been addressed

2. Is the manuscript technically sound, and do the data support the conclusions?

Reviewer #1: Yes

Reviewer #2: Yes

3. Has the statistical analysis been performed appropriately and rigorously? 

Reviewer #1: Yes

Reviewer #2: Yes

4. Have the authors made all data underlying the findings in their manuscript fully available?

Reviewer #1: Yes

Reviewer #2: Yes

5. Is the manuscript presented in an intelligible fashion and written in standard English?

Reviewer #1: Yes

Reviewer #2: Yes

6. Review Comments to the Author

Reviewer #1: Thank you for your kind correction.

I am satisfied with the revised manuscript. I have no further comments. Congratulations!!

Reviewer #2: All the points have been addressed by the authors. I kindly ask the editor to accept the mansucript.

7. PLOS authors have the option to publish the peer review history of their article (what does this mean?). If published, this will include your full peer review and any attached files.

Reviewer #1: No

Reviewer #2: No

---

## [Editor Report · Acceptance letter]

20 Sep 2021

PONE-D-21-19762R1 

Reliability of ultrasonographic measurement of muscle architecture of the gastrocnemius medialis and gastrocnemius lateralis 

Dear Dr. Kingsley:

I'm pleased to inform you that your manuscript has been deemed suitable for publication in PLOS ONE. Congratulations! Your manuscript is now with our production department. 

Kind regards, 

on behalf of

Professor Emiliano Cè 

Academic Editor

PLOS ONE